# Pathway connectivity and signaling coordination in the yeast stress-activated signaling network

Deborah Chasman[1,†], Yi-Hsuan Ho[2,†], David B Berry[2,‡], Corey M Nemec[3], Matthew E MacGilvray[2], James Hose[2], Anna E Merrill[4], M Violet Lee[4,§], Jessica L Will[2,¶], Joshua J Coon[4,5,6], Aseem Z Ansari[3,5,*], Mark Craven[1,5,7,**] & Audrey P Gasch[2,5,***]

## Abstract

Stressed cells coordinate a multi-faceted response spanning many levels of physiology. Yet knowledge of the complete stress-activated regulatory network as well as design principles for signal integration remains incomplete. We developed an experimental and computational approach to integrate available protein inter-action data with gene fitness contributions, mutant transcriptome profiles, and phospho-proteome changes in cells responding to salt stress, to infer the salt-responsive signaling network in yeast. The inferred subnetwork presented many novel predictions by impli-cating new regulators, uncovering unrecognized crosstalk between known pathways, and pointing to previously unknown 'hubs' of signal integration. We exploited these predictions to show that Cdc14 phosphatase is a central hub in the network and that modi-fication of RNA polymerase II coordinates induction of stress-defense genes with reduction of growth-related transcripts. We find that the orthologous human network is enriched for cancer-causing genes, underscoring the importance of the subnetwork's predictions in understanding stress biology.

**Keywords** environmental stress; integer programming; proteomics; signal transduction; transcriptomics
**Subject Categories** Signal Transduction; Computational Biology; Genome-Scale & Integrative Biology
**Mol Syst Biol. (2014) 10: 759**

## Introduction

All cells respond to stress by orchestrating complex responses customized for each situation. When grown in optimal conditions, *Saccharomyces cerevisiae* maintains high expression of growth-related genes and low transcription of stress-defense genes, in part via nutrient responsive TOR and RAS-regulated protein kinase A (PKA) signaling (Smets *et al*, 2010; Broach, 2012). Suboptimal conditions suppress these pathways in an unknown manner while activating stress-specific signaling networks that coordinate changes in transcription and translation, protein function, and metabolic fluxes with transient arrest of growth and cell cycle progression. How these disparate physiological processes are coordinated is poorly understood but likely critical for surviving and acclimating to stressful conditions.

At the level of gene expression, stressed yeast activate condition-specific transcript changes that provide specialized stress defenses. These responses are typically regulated by condition-specific tran-scription factors (TFs) and upstream signaling pathways that are activated under limited circumstances (Hohmann & Mager, 2003). Concurrently, stressed yeast activate the common environmental stress response (ESR) (Gasch *et al*, 2000; Causton *et al*, 2001). The ESR includes ~300 induced ESR (iESR) genes that are broadly involved in stress defense and ~600 repressed ESR (rESR) genes that together encode ribosomal proteins (RPs) and proteins involved in ribosome biogenesis/protein synthesis (RiBi). While the complete set of ESR regulators remains elusive, it is clear that the program is regulated by different upstream signaling factors under different situations (Gasch *et al*, 2000, 2001; Gasch, 2002). Activation of the ESR, and of transcript changes more broadly, is in fact not required to survive the initial stressor, but rather is necessary for acquired

1  Department of Computer Sciences, University of Wisconsin-Madison, Madison, WI, USA
2  Laboratory of Genetics, University of Wisconsin-Madison, Madison, WI, USA
3  Department of Biochemistry, University of Wisconsin-Madison, Madison, WI, USA
4  Department of Chemistry, University of Wisconsin-Madison, Madison, WI, USA
5  Genome Center of Wisconsin, University of Wisconsin-Madison, Madison, WI, USA
6  Department of Biological Chemistry, University of Wisconsin-Madison, Madison, WI, USA
7  Department of Biostatistics and Medical Informatics, University of Wisconsin-Madison, Madison, WI, USA
*Corresponding author. Tel: +1 608 265 4690; E-mail: ansari@biochem.wisc.edu
**Corresponding author. Tel: +1 608 265 6181; E-mail: craven@biostat.wisc.edu
***Corresponding author. Tel: +1 608 265 0859; E-mail: agasch@wisc.edu
†These authors share first authorship
‡Present address: Institute for Neurodegenerative Disease, University of California-San Francisco, San Francisco, CA, USA
§Present address: Genentech, South San Francisco, CA, USA
¶Present address: University of Georgia, Athens, GA, USA

resistance to subsequent stress (Berry & Gasch, 2008; Westfall *et al*, 2008; Mitchell *et al*, 2009; Berry *et al*, 2011). Therefore, screens for mutants sensitive to a single dose of stress have likely missed many signaling proteins, rendering stress-dependent signaling networks incomplete. Although several isolated 'pathways' are well characterized, how signaling is integrated through a single cellular system is poorly understood.

Here, we present an experimental and computational pipeline to infer the complete sodium chloride (NaCl)-activated signaling network from a combination of data types. A key feature of our approach is that we generated several large-scale datasets (including mutant transcriptome profiles, phospho-proteome changes, and gene fitness contributions) under the same culture system in cells responding to acute NaCl stress. Because stress responses are highly context dependent (Van Wuytswinkel *et al*, 2000; O'Rourke & Herskowitz, 2004; Berry & Gasch, 2008), we restrict our analysis to datasets generated in our own laboratory, despite many insightful prior studies characterizing the salt response in yeast (e.g., Causton *et al*, 2001; Hirasawa *et al*, 2006; Capaldi *et al*, 2008; Melamed *et al*, 2008; Westfall *et al*, 2008; Halbeisen & Gerber, 2009; Soufi *et al*, 2009; Martinez-Montanes *et al*, 2010; Warringer *et al*, 2010; Miller *et al*, 2011).

We wished to develop a computational method to integrate these datasets and infer the stress-activated signaling subnetwork, both to implicate missing regulators and to understand their connections. Prior approaches tackling the challenge of network inference have leveraged large-scale biological datasets, most commonly transcriptome data (see Friedman (2004) and Schadt *et al* (2005)). Extensions focusing on the osmotic response include the work of Gat-Viks *et al*, whose probabilistic method described regulatory relationships between known regulators of the Hog pathway, assuming a known network topology (Gat-Viks *et al*, 2006; Gat-Viks & Shamir, 2007). Several approaches leverage protein–protein and protein–nucleic acid interactions to infer relevant connections between regulators and their downstream gene targets (Liang *et al*, 1998; Ideker *et al*, 2000; Yeang *et al*, 2004; Yeung *et al*, 2004; Markowetz *et al*, 2005; Tu *et al*, 2006; Suthram *et al*, 2008; Huang & Fraenkel, 2009, 2012; Vaske *et al*, 2009; Yeger-Lotem *et al*, 2009; Novershtern *et al*, 2011). The method we present here is most closely related to methods that infer subnetworks by solving an integer linear program (IP) (Ourfali *et al*, 2007; Gitter *et al*, 2011; Silverbush *et al*, 2011). In particular, Gitter *et al* (2013) developed a combined probabilistic/IP method to discern signaling in the potassium chloride-responsive subnetwork from time series expression data (Gitter *et al*, 2013). However, their approach incorporated transcriptome data only, whereas we were interested in incorporating other data types. Methods that integrate disparate datasets are emerging, for example, the work of Huang *et al* (2013) that considered existing transcriptomic and proteomic data to study oncogene-induced signaling (Huang *et al*, 2013). In our case, we wanted to design a method that could also take mutant transcriptome profiles generated in our own laboratory.

We therefore designed an integer linear programming (IP) approach to integrate and interpret our disparate datasets by inferring a signaling subnetwork. The novel facets of our computational approach include a means to integrate these varied data sources, using new types of input paths to the IP, and a multi-part objective function. The resulting subnetwork generated many new insights

into stress signaling, by implicating new regulators, unveiling the connections between them, and presenting organization principles that shed light on stress biology.

## Results

We previously identified 225 genes important for acquired stress resistance after NaCl pretreatment (Berry *et al*, 2011), including a subset of the known signaling proteins activated by NaCl (Supplementary Fig S1). Because only a fraction of NaCl-dependent transcript changes are important for acquired stress resistance, the selection misses many of the upstream transcriptome regulators. Therefore, to implicate the complete upstream signaling subnetwork, we began by profiling NaCl-dependent expression changes in 16 mutants implicated in NaCl-induced acquired stress tolerance (Fig 1, see Materials and Methods). Together, this generated a matrix of regulator–gene target predictions that encompassed 3,300 genes (Supplementary Fig S2 and Table 1). A third of the affected genes were dependent on ≥ 2 regulators, and there was significant overlap in several target-gene sets (hypergeometric test, Fig 1). These results hint at the complex upstream signaling that controls the NaCl-responsive transcriptome.

Because much of signal transduction occurs post-translationally, we next measured changes to the phospho-proteome before and at 5 and 15 min after NaCl treatment, using chemical isobaric tags for phosphopeptide quantification (see Materials and Methods). Nearly 600 of 1,937 identified phospho-sites (mapping to 973 proteins) showed a ≥ 2-fold change in phosphorylation, roughly split between

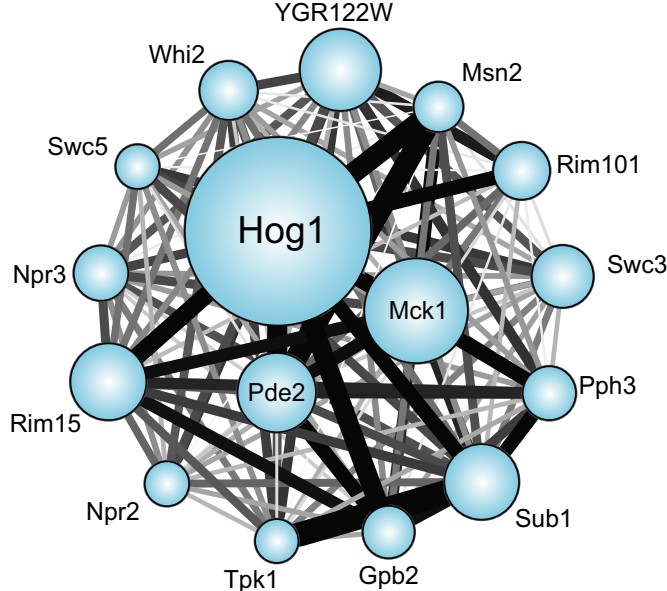

**Figure 1. Overlapping targets of interrogated 'source' regulators.**
The number of genes whose osmotic response was defective in each of 16 mutants is represented by the size of each circle. Edge thickness represents the fraction of the smaller node's targets that overlap between two nodes. Edge color is proportional to significance of the overlap (hypergeometric test), where black represents a −log(*P*-value) of 5 or greater.

**Table 1. Gene targets identified in regulator mutants.**

| Mutant[a] | Defective[b] | Amplified[b] |
|---|---|---|
| Source regulators | | |
| hog1Δ (3) | 1378 | 565 |
| pde2Δ (3) | 517 | 59 |
| mck1Δ (3) | 794 | 101 |
| msn2Δ (3) | 184 | 26 |
| rim101Δ (3) | 75 | 227 |
| gpb2Δ (2) | 202 | 37 |
| rim15Δ (2) | 438 | 106 |
| npr2Δ (2) | 75 | 69 |
| npr3Δ (2) | 184 | 89 |
| swc3Δ (2) | 108 | 257 |
| swc5Δ (2) | 84 | 55 |
| whi2Δ (2) | 118 | 201 |
| pph3Δ (2) | 235 | 21 |
| sub1Δ (2) | 431 | 97 |
| tpk1Δ (2) | 35 | 96 |
| ygr122wΔ (2) | 106 | 502 |
| Validation mutants | | |
| cdc14-3 (3)[c] | 929 | 346 |
| nnk1Δ (1) | 94 | 278 |
| bck1Δ (1) | 107 | 169 |
| yak1Δ (1) | 226 | 248 |
| kin2Δ (1) | 52 | 266 |
| pho85Δ (1) | 614 | 342 |
| cka2Δ (2) | 155 | 63 |
| cka1Δ (2) | 58 | 133 |
| ckb1Δ ckb12Δ (2) | 129 | 176 |
| arf3Δ (2) | 466 | 331 |
| scd6Δ (2) | 0 | 0 |

[a]Mutant and number of replicates in parentheses.
[b]Number of genes with smaller (defective) or larger (amplified) expression changes compared to the wild-type strain. Note, this table includes non-coding RNAs that were excluded from the inference. The table lists the number of targets identified from the originally interrogated 'source' regulators and validation mutants.
[c]cdc14-3 was compared to its isogenic and identically treated wild-type.

sites with increased and decreased modification (Supplementary Fig S3). Over 10% of the altered phospho-proteins represented kinases and phosphatases (including regulators of cell cycle progression, actin organization, and signal transduction) as well as transcriptional regulators (such as activators Hot1, Sko1, and Sub1 and repressors Mot2, Dot6, and Dig1). Proteins affected at the later time point were involved in cytokinesis, bud-site selection, and actin reorganization (Bonferroni-corrected $P < 0.01$, hypergeometric test), implying downstream physiological effects on these processes.

This analysis generated a rich source of datasets (outlined in Fig 2). To integrate and interpret these disparate datasets, we designed an integer linear programming-based (IP) approach (Fig 3 and Materials and Methods). Using a *background network* of

physical or chemical protein interactions, the method infers a *subnetwork* that predicts the *paths* by which each interrogated *source* regulator is connected to its downstream *targets* (identified as dysregulated genes in the *source* mutant responding to NaCl treatment). Each path is a directed, linear chain of interactions between yeast proteins, where the terminal protein node represents a sequence-specific transcription factor (TF) or RNA-binding protein (RBP) known to bind the downstream promoters or transcripts, respectively. The IP's objective function favors the inclusion of salt-responsive proteins, that is, those with differential phosphorylation or required for acquired stress fitness after NaCl treatment, and allows the sparing inclusion of additional proteins.

Specifically, we start with a background network of directed and undirected intracellular interactions representing protein–protein, kinase–substrate, and gene regulatory interactions between proteins and genes/mRNAs (Guelzim *et al*, 2002; Ptacek *et al*, 2005; MacIsaac *et al*, 2006; Stark *et al*, 2006; Hogan *et al*, 2008; Everett *et al*, 2009; Pu *et al*, 2009; Breitkreutz *et al*, 2010; Scherrer *et al*, 2010; Tsvetanova *et al*, 2010; Abdulrehman *et al*, 2011; Fasolo *et al*, 2011; Sharifpoor *et al*, 2011; Venters *et al*, 2011; Heavner *et al*, 2012; Huebert *et al*, 2012). For each interrogated source regulator, we identify candidate TFs and RBPs whose known binding targets significantly overlap with the source's targets (Fig 3A). We then enumerate all possible directed candidate paths (using an iterative deepening search up to a given length) that connect each of the 16 interrogated *source* regulators to the majority of their *targets*, through candidate TFs or RBPs (Fig 3B). Other candidate paths connect proteins required for fitness (Fig 3B, blue nodes), proteins with NaCl-dependent phosphorylation changes (yellow nodes), and two known upstream sensors (pink nodes). The candidate paths serve as input to the IP, which encodes the relevance of each network element as a binary variable and characterizes possible subnetworks using a set of linear constraints over these variables (Fig 3C). Subnetwork inference is performed by choosing a union of relevant, directed paths that optimize a series of successively applied objective functions that aim to connect experimentally implicated proteins while minimally including proteins not currently supported by experimental evidence. Because many distinct subnetworks may score equally well, we use the IP to identify an ensemble of high-scoring subnetworks. In turn, each protein, interaction, and path is assigned a confidence value based on its frequency across the ensemble.

### Validation analysis provides strong support for the inferred subnetwork

Using the datasets described above, the method identified a consensus subnetwork encompassing 380 nodes (predicted regulators) and 1,131 edges (relevant interactions) present at 75% confidence (Fig 4A). To assess the inferred subnetwork's predictive accuracy, we performed precision–recall analysis using an assembled list of known NaCl regulators and another list of unlikely regulators that included metabolic enzymes and exclusively subcellular proteins. We excluded from consideration proteins with phospho-changes or fitness contributions (since they are preferentially included by the inference) and plotted the precision and recall over varying node-confidence thresholds (Fig 4B). The inferred ensemble achieved substantially higher accuracy than the

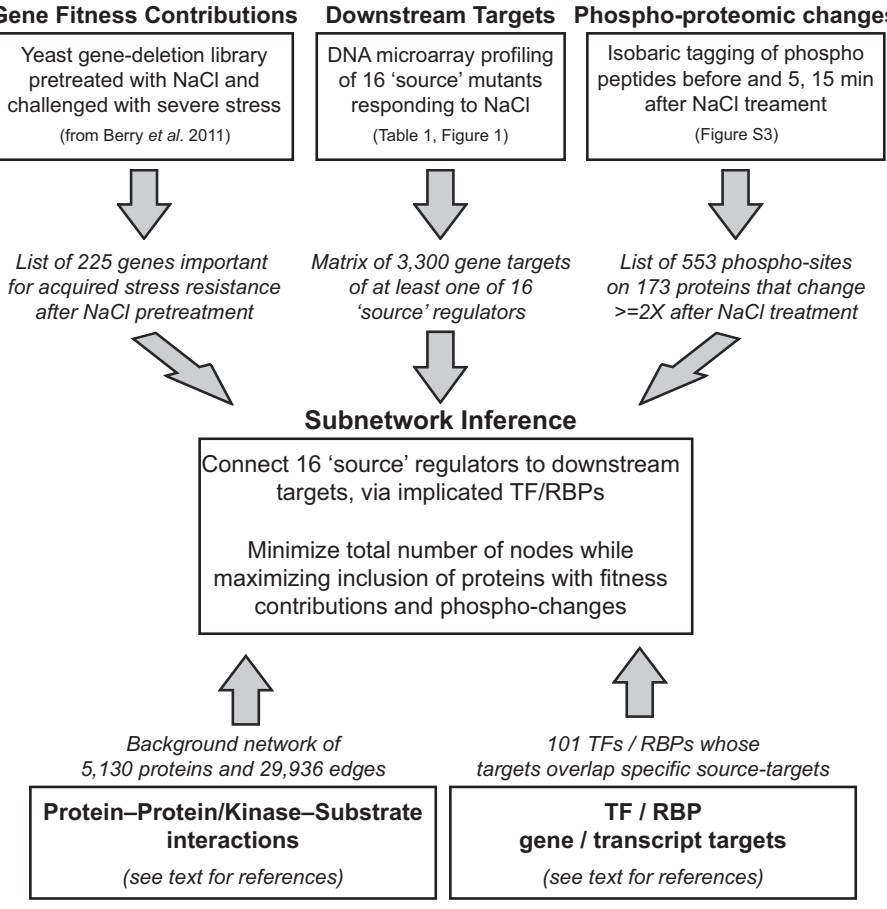

**Figure 2. Overview of the experimental data collection and analysis to generate IP input.**
See text for details.

enumerated candidate paths provided as input to the IP method, highlighting the power of the inference step (Fig 4B, green line). To assess the effects of the topological properties in the background network, we ran the method on permuted source–target pairs (maintaining the degree distribution from the real data; see Materials and Methods). This permuted baseline achieved high accuracy in the low-recall range, suggesting that some regulators are highly central in the background network. However, our inferred ensemble significantly outperformed the permuted baseline at higher levels of recall; thus, our method's accuracy is not simply due to properties of the background network's topology. To understand the contribution of each component of our method, we also performed additional enrichment analyses and other computational evaluations, with results available in Supplementary Information Section 2.

We found additional support for the inferred subnetwork in the non-random inclusion of specific protein functional groups. When compared to the background network, to the enumerated candidate pathways used as input to the IP, and to the permuted subnetworks, the inferred consensus subnetwork was enriched for proteins annotated as 'stress' proteins (background, $P = 5e-21$; candidates, $P = 2e-6$; permutations, $P = 0.007$) and for proteins encoded by genes with genetic interactions (background and candidates, $P \approx 0$; permutations, $P = 0.003$) (Stark *et al*, 2006), which suggests

functional dependencies. The consensus subnetwork was also slightly enriched for kinases (relative to the candidate paths and background network) and for essential genes (relative to the background network), but not relative to the permuted subnetworks (suggesting its bias toward kinases and essential genes).

The inferred subnetwork included many regulators not previously linked to the NaCl response. To test some of the novel predictions, we analyzed osmo-dependent transcriptome changes in 14 mutants lacking predicted regulators, with preferences for kinases and phosphatases (see Table 1; Supplementary Fig S4 and Supplementary Information). The results provided strong support overall for the inferred subnetwork. All but one of the mutants (93%) displayed a defect in osmo-responsive expression. Furthermore, the predicted targets of 80% of these regulators overlapped significantly ($P < 1e-3$) with their measured targets, highlighting the accuracy of regulator–target predictions. To garner support for the subnetwork's structure, we investigated the overlap in targets of each interrogated mutant and the known or measured targets of proteins predicted to lie in the interrogated regulator's paths. Using stringent scoring, we found support for 30–100% of nodes in most paths (53% on average, Supplementary Table S1). Together, these results provide strong support for the validity of the inferred consensus subnetwork.

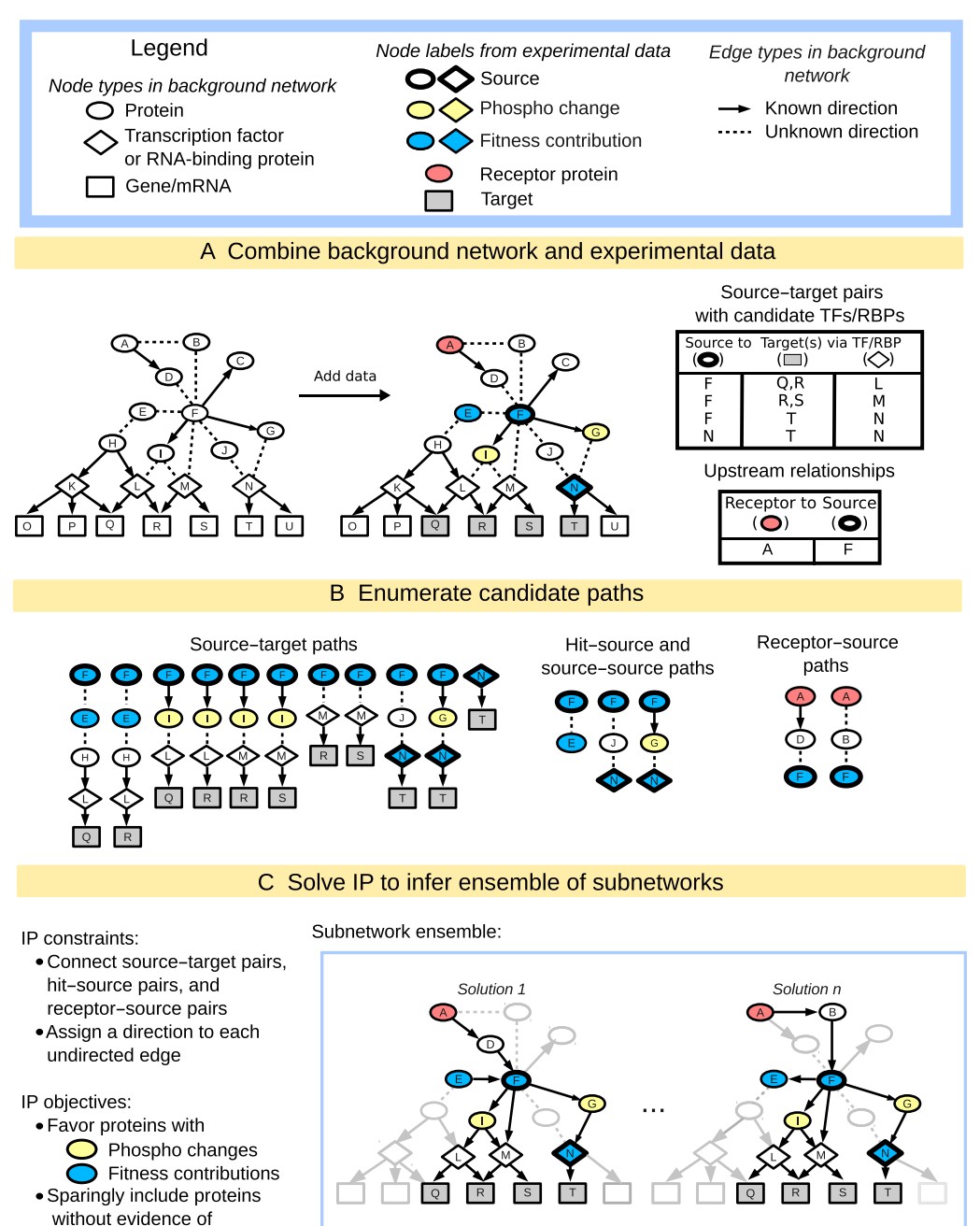

**Figure 3.  Overview of the subnetwork inference method.**

A  The input to the method includes a background network of yeast interactions combined with experimental data that describes the yeast salt stress response, including proteins with phospho-changes (yellow), fitness contribution (blue), or two known upstream regulators (pink), as described in the key.

B  The three different types of paths that we enumerate using the background network and experimental data, where 'hit' refers to proteins identified in the original fitness screen or with significant changes in phosphorylation.

C  The IP for subnetwork inference and the output ensemble of inferred subnetworks.

## Known and new players captured in the NaCl-responsive signaling subnetwork

We therefore explored the consensus subnetwork for new insights into stress signaling. Many expected pathways were captured, including the canonical HOG, PKA, and TOR pathways. The inferred subnetwork included other stress-activated pathways not previously linked to the NaCl response, such as PKC, Pho85, Rim15 pathways, and GSK-3 kinase Mck1 (Fig 5A). We tested the involvement of these pathways by analyzing our phospho-proteomic data and mutant transcriptome profiles: We found that members of all of these pathways showed NaCl-dependent phospho-changes, and cells

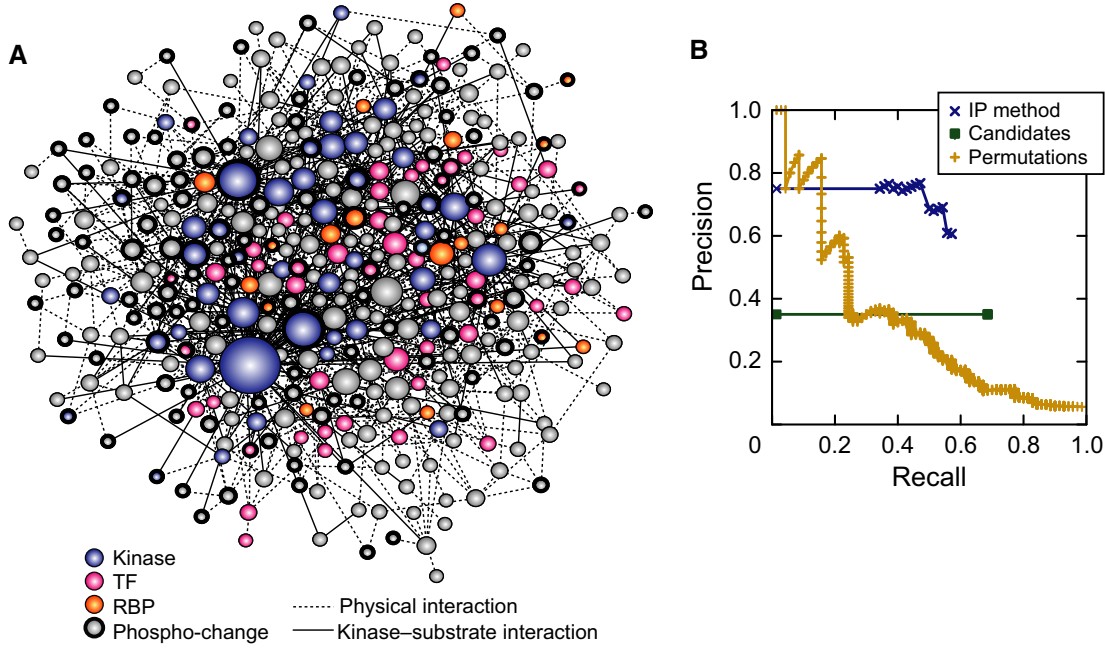

**Figure 4. Inferred NaCl-activated signaling network.**

A  Inferred consensus subnetwork at 75% confidence, where node size indicates degree (number of connections) and color is according to the key. Nodes representing proteins with phospho-changes are outlined in bold.

B  Precision–recall of the inferred consensus network was calculated using a list of true positives from the literature and a list of likely negatives, after excluding proteins with phospho-changes and those required for fitness (see Materials and Methods). *Precision* is the fraction of predicted nodes known to be involved in the osmo response, and *recall* is the fraction of true positives that are above the threshold. The curves represent the performance of the IP method on the real data (blue), of the method on randomized permutations of the input network (yellow), and of the candidate enumerated pathways used as input (green, see Materials and Methods).

lacking specific pathway members (including *BCK1*, *YAK1*, *PHO85*, *RIM15*, and *MCK1*) had defects in NaCl-dependent expression changes (Supplementary Fig S4 and Supplementary Information). The subnetwork also included the 'STE' mating pathway, which shares upstream components with the Hog network and is known to be suppressed by Hog1 signaling (O'Rourke & Herskowitz, 1998; Marles *et al*, 2004; Zarrinpar *et al*, 2004; McClean *et al*, 2007; Shock *et al*, 2009; Patterson *et al*, 2010; Nagiec & Dohlman, 2012). The inclusion of the mating pathway indicates that some connections in the consensus subnetwork represent signaling suppression that prevents crosstalk to other pathways. We also validated several newly implicated regulators, including the CK2 kinase complex (see

Supplementary Information) and the Cdc14 phosphatase (see below).

## Interconnectivity in the inferred signaling subnetwork

The structure of the subnetwork revealed surprising cross-connectivity between previously defined pathways. We defined stress-activated 'pathways' based on the literature and then summed the number of direct connections between members of those pathways (Fig 5B). Many of the pathways were intricately connected, with Tor1 and PKA pathways linked to the greatest number of other pathways. We also identified individual subnetwork nodes as

**Figure 5. Connectivity between known pathways and hubs of signal integration.**

A  A subregion of the inferred subnetwork, highlighting proteins in known pathways according to the key. Hexagons represent interrogated 'source' regulators, nodes outlined in bold indicate validated players in the NaCl response, and asterisks represent proteins with phospho-changes upon NaCl treatment. Dashed edges represent physical interactions and solid arrows indicate kinase–substrate relationships. Edge directionality is as predicted by the inference, and edge color is according to the edge's source node. Inhibitory edges were taken from the literature.

B  Connectivity between known pathways, where blue boxes represent the number of interactions between any members of two pathways. Pathway membership is indicated in parentheses.

C  The top 15-ranked 'integrator' nodes with connections to the greatest number of different pathways, as shown in (B).

D  A purified CTD peptide was incubated with Hrr25-TAP or Hog1-TAP purified from cells with and without NaCl treatment for 10 min, incubated with and without the reversible p38-specific inhibitor SB203580 (INH) added *in vitro*. Reactions with buffer or yeast whole-cell extract (WCE) served as negative and positive controls, respectively. CTD phosphorylated on serine 2 (Ser2) or Ser5 was detected by immunoblotting (see Materials and Methods). TAP-tagged proteins were subsequently quantified on the same blot with the anti-TAP antibody. Quantification of Hog1 phosphorylation, shown to the right, was normalized to Hog1-TAP abundance and then to the corresponding unstressed sample.

'integration' points, defined as nodes with the greatest number of connections to distinct pathways (Fig 5C). Nearly half of the top ten integration nodes were kinases or phosphatases, including Mck1

and cell cycle regulator Cdc28, which regulates RP genes under optimal conditions (Chymkowitch *et al*, 2012) but is suppressed during osmotic shock (Alexander *et al*, 2001; Belli *et al*, 2001; Adrover

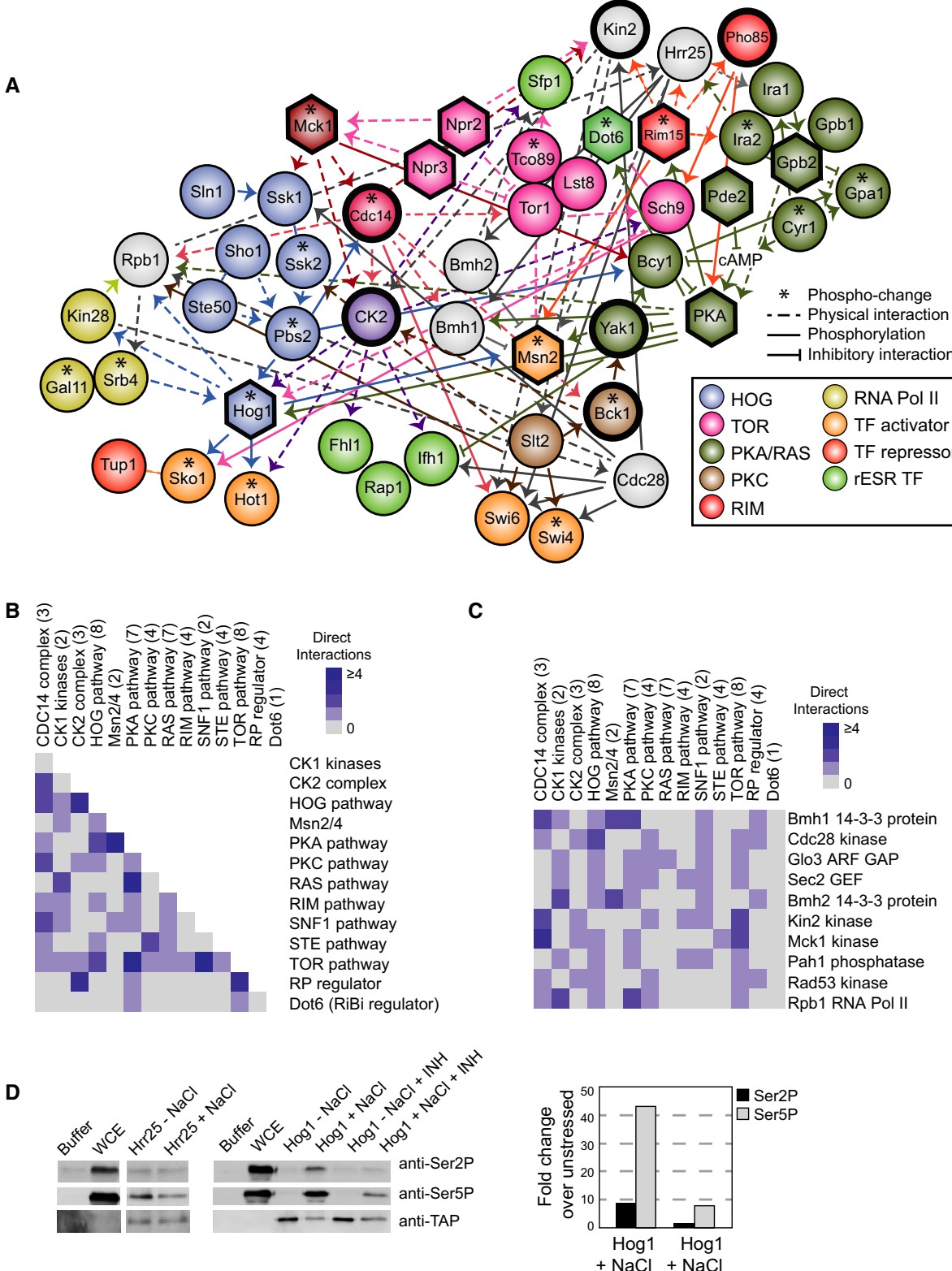

**Figure 5.**

*et al*, 2011). 14-3-3 proteins Bmh1 and Bmh2 were also identified as integration points, confirming their known role as signaling cofactors.

Several of the integration points are also hubs of high connectivity in the consensus subnetwork. While 11 of the top 15 most connected nodes are kinases or phosphatases, the remaining four are known regulatory cofactors—including stress-activated ubiquitin (Ubi4), Sumo (Smt3), and Bmh1—and the core subunit of RNA polymerase (Pol II), Rpb1. Modification of the Rpb1 carboxyl-terminal domain (CTD) is the basis for the so-called CTD code of transcriptional regulation (Buratowski, 2003; Zhang *et al*, 2012), making it a logical downstream integration point for complex upstream signaling. Consistent with the predictions of the subnetwork, we found that two of the Rpb1-interacting kinases—Hrr25 and Hog1—phosphorylate the Rpb1-CTD *in vitro*. TAP-tag-purified Hrr25-TAP phosphorylated Rpb1-CTD serine 5 (Ser5), regardless of prior NaCl treatment (Fig 5D). In contrast, TAP-purified Hog1-TAP phosphorylated both Ser2 and Ser5, but only after cellular NaCl treatment and in a manner inhibited by a Hog1-specific inhibitor added *in vitro*. Both Hrr25 and Hog1 are known to interact with Pol II and influence transcriptional processes (Alepuz *et al*, 2003; Phatnani *et al*, 2004; Proft *et al*, 2006; Cook & O'Shea, 2012; Nadal-Ribelles *et al*, 2012) but neither had been implicated in direct Rpb1-CTD phosphorylation. These results are consistent with the model that the Rpb1-CTD is a direct target of the signaling network and plays a central role in signaling (see more below).

We also dissected the regulatory connections surrounding a second hub, Cdc14. In the process, we found that Cdc14 is critical for coordinating distinct facets of the NaCl response. First, the defect in NaCl transcriptome changes evident in *cdc14-3* cells overlapped significantly with the Hog1 response, raising the possibility that Cdc14 is important for Hog1 regulation (Supplementary Fig S4). The subnetwork predicts that Cdc14 is activated in part by the Hog1 regulator Pbs2 (reminiscent of Pbs2 control of Cdc14 localization during the cell cycle (Reiser *et al*, 2006)) and that Cdc14 affects Hog1 function via the nuclear exporter, Crm1. This prompted us to follow Hog1 localization in the *cdc14-3* mutant. Indeed, Hog1 nuclear localization was defective in the NaCl-treated *cdc14-3* mutant (Fig 6A; Supplementary Information), despite Hog1 hyper-phosphorylation under these conditions (Fig 6B). We found no direct interaction between Cdc14 and Hog1, suggesting that the hyper-phosphorylation of Hog1 is a secondary response to the defect in nuclear localization rather than a deficit of direct Hog1 dephosphorylation by Cdc14. The aberrant Hog1 localization was not a side effect of cell cycle arrest, since we found no defect in wild-type cells progressing through G2/M phase or in nocodazole-arrested cells (Supplementary Fig S6). Instead, these results suggest a direct connection between Cdc14 activity and signaling through the Hog pathway.

Second, the subnetwork predicts that Cdc14 regulates CK2 subunits to modulate the Hog1-regulated TF, Hot1. We uncovered a salt-enhanced interaction between Cdc14 and CK2 subunit Cka2 (Fig 6C) and uncovered a constitutive association between CK2 subunits Cka1/Cka2 and Hog1 (Fig 6C and D). Although the connection between CK2 and Hog1 was not known in yeast, our results are reminiscent of regulation in mammalian systems, in which CK2 is regulated by the human ortholog of Hog1, p38 (Sayed *et al*, 2000; Hildesheim *et al*, 2005; De Amicis *et al*, 2011; Isaeva & Mitev, 2011). As predicted by the subnetwork, we found that Cdc14, Cka2, and Hog1 were all required for normal induction of Hot1 targets (Fig 6E).

Finally, and surprisingly, we discovered that Cdc14 suppresses NaCl-dependent crosstalk to the cell cycle network: the *cdc14-3* mutant at the non-permissive temperature strongly and aberrantly induced G1 and S phase genes upon NaCl treatment (Fig 6F), even though cells were completely arrested in M phase for the duration of the treatment. This included genes encoding G1 and S phase cyclins *CLN1/2* and *CLB5/6*, respectively. To further understand this effect, we turned to the subnetwork: Cdc14 is predicted to affect these genes via direct interaction with the carbon-responsive kinase Snf1, which is known to be activated by NaCl (Hong & Carlson, 2007; Ye *et al*, 2008). Snf1 is also required for proper timing of cell cycle entry in standard conditions (Pessina *et al*, 2010; Busnelli *et al*, 2013), raising the possibility that it is responsible for the inappropriate G1/S gene induction in the absence of Cdc14. We found that deletion of *SNF1* in the *cdc14-3* background largely abrogated the hyper-activation of G1 and S genes in the *cdc14-3* mutant (Fig 6G). This presents a model for future dissection, in which Cdc14 helps to suppress the cell cycle effect of Snf1 activation, thereby funneling Snf1 activity toward its stress-specific gene targets. Together, our results demonstrate the remarkable and central role of Cdc14 in coordinating cellular signaling upon osmotic shock, while showcasing the predictive power of our inferred subnetwork.

## New insights into ESR regulation and coordination

We were especially interested in how distinct modules in the ESR—including iESR genes important for stress defense and RP/RiBi modules required for rapid growth—are regulated and coordinated. Of the 178 nodes implicated in ESR regulation, over half were predicted (Fig 7A)—and several confirmed (Fig 7B)—to lie upstream of all three ESR modules. In contrast to common upstream nodes that were enriched for kinases compared to the consensus subnetwork ($P = 2.6e-7$), nodes exclusive to iESR regulation were enriched for TFs ($P = 5e-5$), while rESR regulators showed a preponderance of RBPs ($P = 1e-5$), implicating regulated RNA stability for these genes. Many more regulators and regulatory connections were unique to the iESR versus RP and RiBi modules (the latter being the largest group) (Fig 7C). This is consistent with the extensive redundancy in iESR control (Gasch, 2002) and hints at a more monolithic regulation of rESR expression during times of adversity.

To better understand how cells coordinate repression of growth-related genes with induction of stress-defense genes in the ESR, we devised a bifurcation score based on information theory, to rank nodes that (a) are upstream of many genes from both modules but (b) have outgoing paths that relatively cleanly divide iESR and rESR genes. A third of top 15-ranked bifurcating nodes are linked to cAMP signaling (including adenylate cyclase Cyr1, cAMP response regulator Bcy1, and phosphodiesterase Pde2). Indeed, we found that the *pde2Δ* mutant has a defect in both iESR induction and rESR repression (Fig 7B), confirming the role of cAMP in the growth/stress-defense decision (see Discussion). Nearly half of the remaining top-ranked bifurcation proteins associate with RNA Pol II (including Pol II core subunit Rpb3, Pol II-associated Sub1 and Ask10, transcription elongation factor Spt5, as well as Sds3 of the Rpd3L chromatin remodeling complex). Together with the

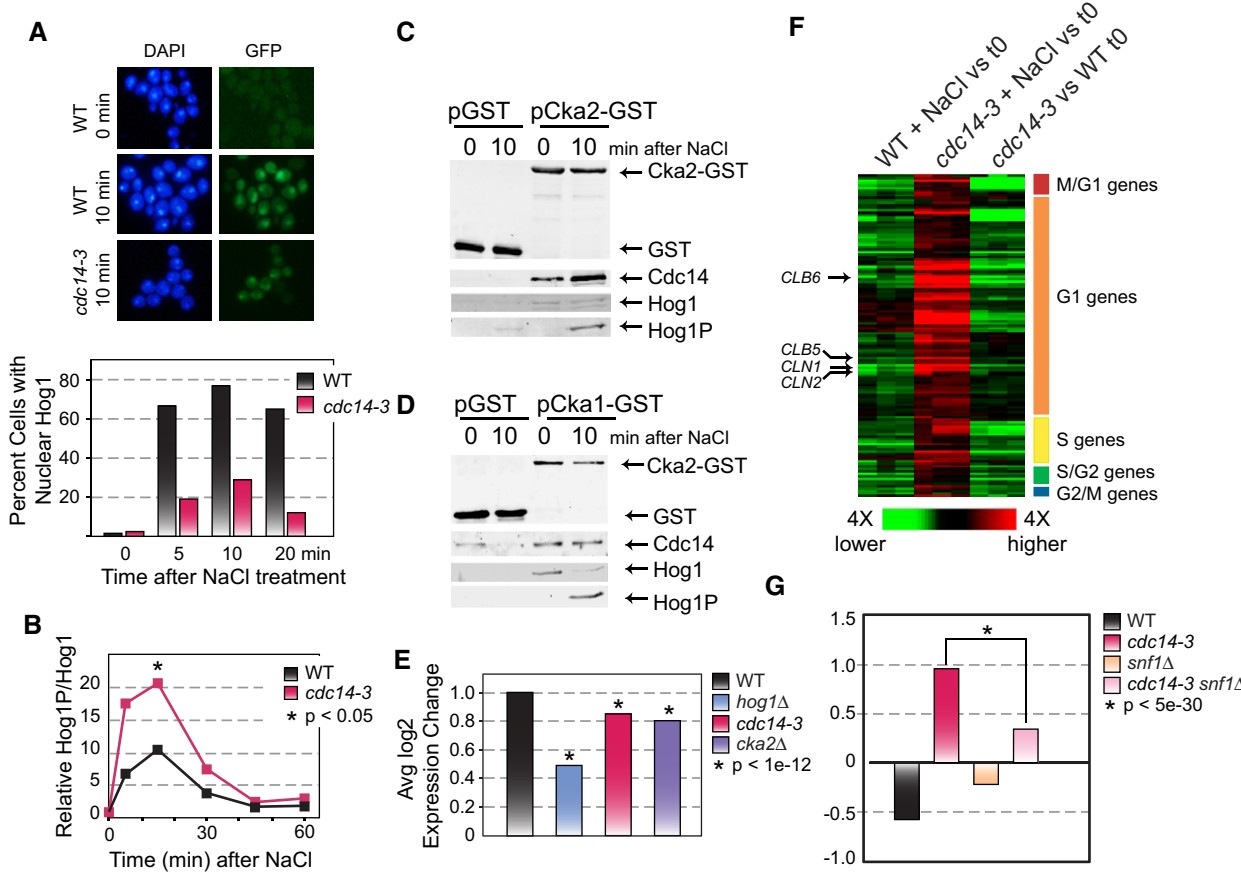

**Figure 6. Cdc14 is a central regulator in the NaCl response.**

A   Wild-type (WT) and *cdc14-3* cells were shifted to 35°C for 90 min and then exposed to 0.7 M NaCl for up to 20 min. Images represent nuclear DNA (DAPI, left) and Hog1-GFP (right) before and at 10 min after NaCl treatment. The plot below quantifies the fraction of cells ($n \geq 75$) with nuclear Hog1-GFP signal that overlapped the DAPI signal, in WT and *cdc14-3* cells.

B   Levels of phospho-Hog1 normalized to total Hog1 in WT and *cdc14-3* cells responding to NaCl at 35°C. Data represent the average of biological duplicates (paired *t*-test).

C, D   GST-tagged Cka2 (C) or GST-tagged Cka1 (D) were immunoprecipitated and blotted for Cdc14 and total or phospho-Hog1.

E   The average $\log_2$ fold-change of 67 Hot1 targets in replicated WT, *hog1Δ*, *cdc14-3*, and *cka2Δ* strains responding to NaCl. Data for each mutant and its paired WT were scaled to the plotted WT so as to accurately represent the mutant defect. Asterisks represent a significant difference in the mutant versus its paired WT (paired *t*-test).

F   Expression data in WT or *cdc14-3* cells responding to NaCl at the non-permissive temperature and in *cdc14-3* cells versus WT at the non-permissive temperature before NaCl addition. Each column represents one of three triplicated expression responses, and each row represents one of 131 cell cycle genes aberrantly induced in *cdc14-3* after NaCl treatment (FDR < 0.05). Red represents higher and green represents lower expression in response to NaCl (or in the *cdc14-3* mutant in the case of the last columns), according to the key. Cell cycle classification of the genes (Spellman *et al*, 1998) is shown to the right; cyclins are annotated to the left.

G   Average $\log_2$ expression change of genes shown in (F), as described in (E).

Source data are available online for this figure.

---

identification of Pol II subunit Rpb1 as a hub in the subnetwork, these results implicate RNA Pol II at a key decision point in ESR coordination.

To investigate this, we started by checking *in vivo* bulk modification of Rpb1-CTD in wild-type and *hog1Δ* cells responding to NaCl. The *hog1Δ* mutant showed an initial drop in Ser5 and Ser2 phosphorylation similar to the wild-type, but displayed a reproducible defect in the normal subsequent transient increase in Ser5 and Ser2 phosphorylation (Fig 8A). The timing of the transient peaks in bulk Rpb1-CTD phosphorylation correlates with the timing of transcription initiation and elongation upon osmotic stress (Berry & Gasch, 2008; Lee *et al*, 2011; Miller *et al*, 2011), consistent with the known roles of Hog1 as well as Ser5 and Ser2 phosphorylation in

these processes (Alepuz *et al*, 2003; Proft *et al*, 2006; Zhang *et al*, 2012).

To test our hypothesis that direct modification of Rpb1-CTD is important for ESR regulation, we measured transcriptomic changes upon salt stress in Rpb1-CTD mutant strains that could not be phosphorylated normally on CTD-Ser2 or CTD-Ser5 (S2A and S5A mutants, respectively). Since S2A or S5A substitution in all CTD repeats is lethal, the mutant cells expressed chimeric CTD sequences with half mutant and half wild-type repeat sequences (West & Corden, 1995; see Materials and Methods). Neither mutant showed significant expression differences in the absence of stress; furthermore, the S2A mutant showed only a subtle defect in NaCl-dependent expression changes (Fig 8B). In contrast, the S5A mutant

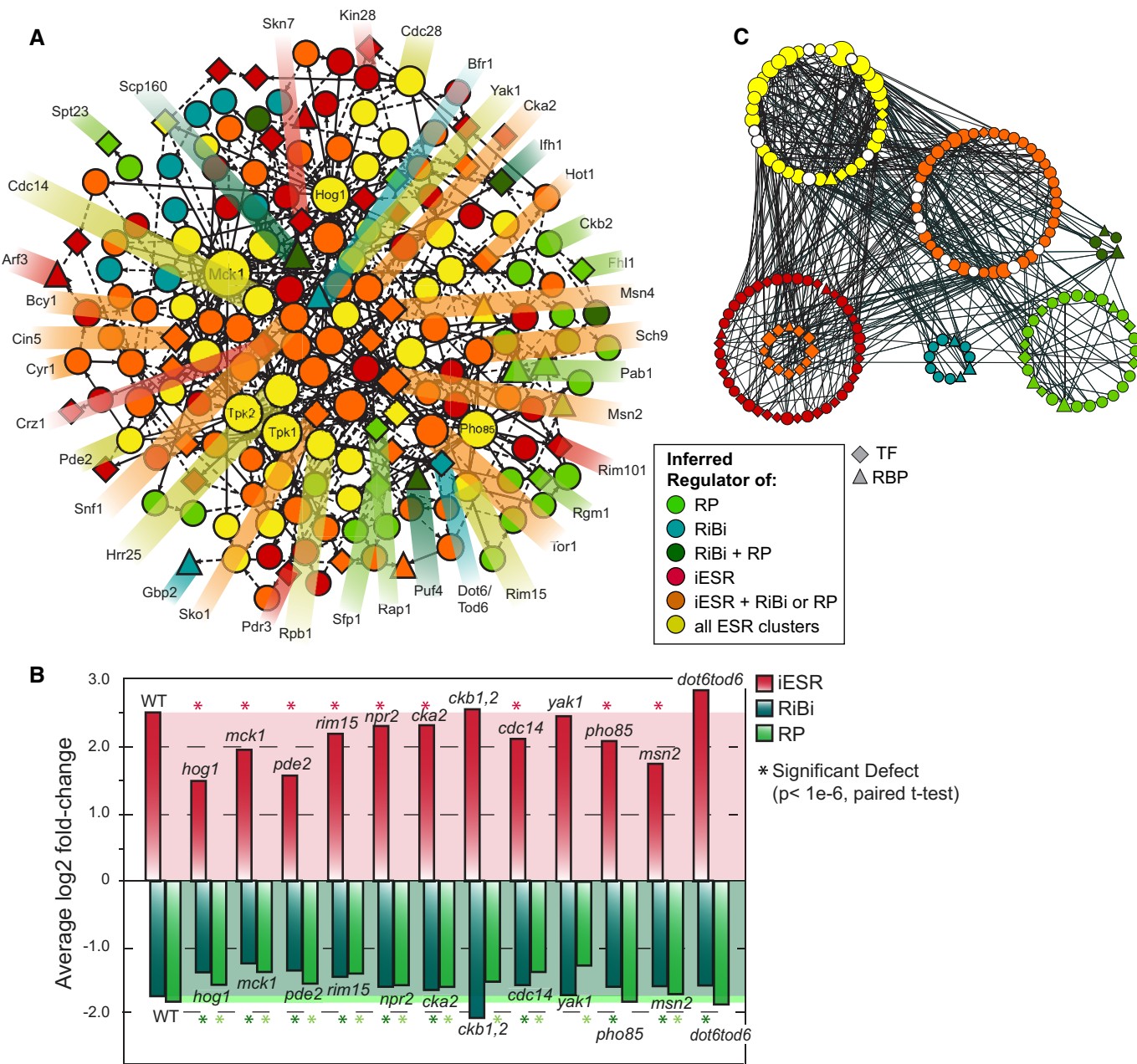

**Figure 7.  Inferred ESR regulatory subnetwork.**

A   Regulators predicted to lie upstream of iESR, RP, and/or RiBi ESR modules are color-coded according to the key and sized according to degree (number of connections). Diamonds and triangles represent TFs and RBPs, respectively.

B   Average log$_2$ expression changes of iESR, RP, or RiBi genes in mutants responding to salt. Wild-type (WT) levels are highlighted by shaded areas; genes with a significant defect ($P$ < 1e-6, paired $t$-test) are indicated with an asterisk.

C   Same as (A) but organized by ESR regulatory potential. Top-ranked bifurcation nodes discussed in the text are colored white. Nested orange nodes represent iESR TFs that are also predicted to lie upstream of RP paths.

had a significant defect in iESR induction and even more so in rESR repression, comparable to the defect seen in the *hog1*Δ mutant.

We reasoned that aberrant ESR coordination may be caused by an inability of polymerase to re-localize from rESR genes, which are highly transcribed before stress, to stress-induced iESR genes. To test this, we measured chromatin occupancy of RNA Pol II subunit Rpb3 in both wild-type and S5A mutant strains responding to NaCl

stress, using ChIP-chip. In wild-type cells responding to stress, Rpb3 occupancy increased at iESR genes but decreased in the body of RP and RiBi genes, with slight accumulation in the promoter regions of specific repressed genes (Fig 8C; Supplementary Fig S7). In contrast, the S5A mutant showed a reproducible defect in Rpb3 recruitment to iESR genes and a concomitant defect in Rpb3 release from rESR genes (Fig 8C; Supplementary Fig S7). These results show that

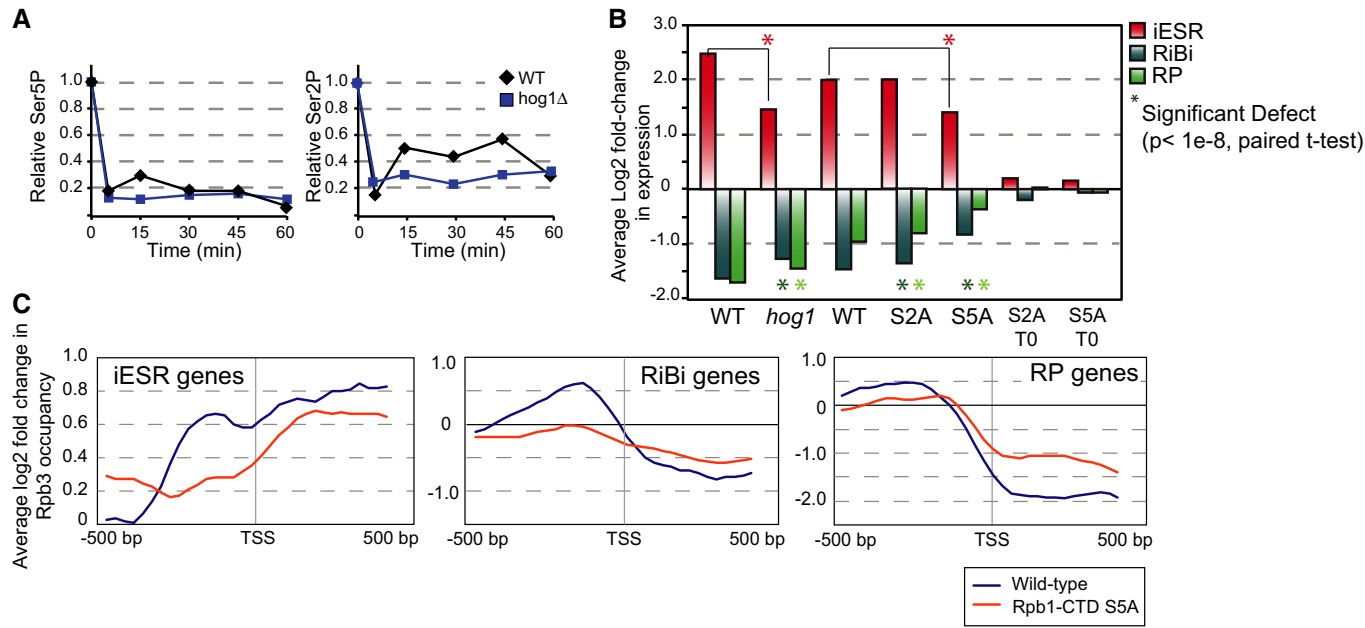

**Figure 8.  Pol II CTD modification coordinates ESR regulation.**

A    Relative abundance of bulk Ser5P (left) and Ser2P (right) normalized to an internal Rpb3 loading control, from yeast cells treated with NaCl for the denoted times. Data are representative of several replicates.

B    The average log$_2$ fold-change expression of iESR, RP, and RiBi genes is shown for paired wild-type and *hog1Δ* strains (as in Fig 7) and paired wild-type, S2A, and S5 strains. There was little expression difference in the S2A and S5A mutants versus wild-type before stress (right bars).

C    Average log$_2$ fold-change in Rpb3 occupancy 20 min after NaCl treatment at ± 500 bp around the transcription start site (TSS) of iESR (left), RiBi (middle), and RP (right) genes. Profiles were reproducible across biological replicates (see Supplementary Fig S7).

direct modification of the Rpb1-CTD is required for normal regulation of iESR and rESR genes (see Discussion).

### The orthologous mammalian networks are enriched for growth-regulating and disease-causing genes

Striking the correct balance between growth rate and stress defense is fundamental for proper cellular function, and improper balance is thought to be a critical driver in diseases such as cancer (Jones & Thompson, 2009). We therefore interrogated the set of human genes orthologous to the yeast NaCl subnetwork. We found that this set is enriched for genes linked to cancer, mostly through somatic mutation, according to the COSMIC database (Forbes *et al*, 2011): of the 35 human genes in the COSMIC dataset with yeast orthologs, 8 were orthologous to nodes in the consensus-node network, representing a 2.5-fold enrichment above chance ($P = 0.0068$, Supplementary Dataset S4A). We also compared the yeast network to Mendelian disease genes in the OMIM database (Hamosh *et al*, 2005). We identified 25 additional yeast genes whose orthologs are linked to heritable disease (Supplementary Dataset S4B), with weak enrichment for genes associated with prostate cancer ($P = 6e-3$, (Woods *et al*, 2013)). The network was also enriched for yeast proteins whose mouse orthologs are required for pre/perinatal viability, normal growth rate and body size, and male and female fertility (FDR < 5%, Supplementary Dataset S5) (Woods *et al*, 2013). These results highlight that stress-responsive signaling is likely important for proper regulation of growth rate, and thus may provide insights into cancer biology (see Discussion).

## Discussion

A major challenge in network biology remains integrating disparate large-scale datasets in a manner that reveals new insights into biology. The approach we developed here provides a new route to identifying the extensive set of players activated during a response, as well as the connections between them and the flow of information toward the processes they regulate.

The computational approach we developed provides several contributions. First, we provide a means to selectively integrate disparate datasets via four types of paths between proteins in the background network. Each dataset is prioritized separately by a series of objective functions, whereas related approaches for inferring signaling networks use a single objective function. One class of related methods essentially maximizes the number of paths between sources and targets (Yeang *et al*, 2004, 2005; Ourfali *et al*, 2007; Gitter *et al*, 2011). In contrast, our method's preference for sparse inferred subnetworks is also employed by approaches based on the prize-collecting Steiner Tree algorithm (Huang & Fraenkel, 2009, 2012; Yosef *et al*, 2009; Huang *et al*, 2013) and flow-based algorithms (Yeger-Lotem *et al*, 2009; Lan *et al*, 2011). However, those methods require the use of a weight parameter to trade off between subnetwork sparsity and the inclusion of known relevant proteins. Another contribution of our approach is the representation of uncertainty in the underlying network. We assign a confidence value to each protein and interaction according to its frequency in an ensemble of optimal inferred subnetworks. This is similar to the score used by Yeang *et al* (2005), who actually enumerate all

optimal solutions; doing so is practically intractable for our input and our model. In contrast, Ourfali et al (2007) assign confidence values based on the change in objective value when each protein or interaction is individually excluded, and Gitter et al (2011) present several methods for ranking paths based on input experimental data and local topological features of the inferred subnetworks. The confidence values generated by our approach provide useful guidance for subsequent biological examination.

The resulting subnetwork put forward by our approach identified new regulators in the NaCl response and provides a glimpse of their connections in a single cellular signaling system. The extensive physical connectivity between what are traditionally considered 'distinct' pathways suggests much greater signaling integration than previously realized. The cross-connectivity between pathways, either direct or through apparent 'integration' points, may coordinate the magnitude or timing of signaling through distinct branches, prevent signaling crosstalk, and/or provide important feedback to dampen signaling as cells acclimate to new conditions (Schwartz & Madhani, 2004; Waltermann & Klipp, 2010). Our results implicate Cdc14 as a critical integrator that bridges HOG and CK2 signaling, suppresses inappropriate activation of the cell cycle network, and connects to several other pathways including Tor1, which is reportedly suppressed by Cdc14 (Breitkreutz et al, 2010). Members of the growth-regulating TOR1 pathway as well as the RAS/PKA pathway show the greatest connectivity to other stress-activated pathways, suggesting that growth regulation is sensitively tuned according to stress conditions.

Our results also shed new light on how growth control and stress defense are related. Optimal growth and maximal stress tolerance are competing interests in the cell: The fastest growing cells are typically the least tolerant of adversity, whereas stress-resistant cells are frequently slow-growing or arrested (Elliott & Futcher, 1993; Sumner & Avery, 2002; Lu et al, 2009; Zakrzewska et al, 2011; Levy et al, 2012). Our results here, along with prior studies, suggest that competition for cellular resources—namely those related to transcription and translation—drive the anti-correlated expression of genes involved in stress defense versus growth promotion. Under optimal growth conditions, rESR transcripts are among the most highly transcribed and the most highly translated (Ingolia et al, 2009; Lipson et al, 2009), consuming the bulk of cellular ribosomes (Warner et al, 2001). We previously proposed that the drop in rESR transcripts helps to direct translational capacity to iESR genes by releasing sequestered ribosomes (Lee et al, 2011). Work by You et al (2013) suggests that cAMP abundance dictates whether translational capacity is directed to growth versus other processes such as stress defense. That our results implicate cAMP in the iESR/rESR regulatory balance is consistent with these models.

In addition to implicating cAMP metabolism, our results show that direct regulation of the RNA Pol II CTD plays a crucial role in the iESR/rESR transcriptional balance, by triggering redistribution of polymerase from highly transcribed growth-related genes to stress-induced defense genes. The ability to fully phosphorylate Ser5 of the Rpb1-CTD is required for normal repression of rESR genes, as indicated by the defect in transcript repression and Pol II redistribution, and is also required for normal recruitment of Rpb3 to iESR promoters for gene induction (Fig 8). Ser5 phosphorylation has been implicated in both gene repression and induction (Hengartner et al, 1998), in support of our findings. The stress-activated redistribution

of Pol II from rESR to iESR genes is at least partly dependent on Hog1 (Cook & O'Shea, 2012; Nadal-Ribelles et al, 2012), which we show phosphorylates the Rpb1-CTD in vitro (Fig 5D) and is required for its normal modification in vivo (Fig 8A). Thus, we propose that direct regulation of RNA Pol II, perhaps in part by the Hog1 kinase, plays a central role in coordinating these opposing transcriptional modules.

Establishing the correct balance between stress tolerance and growth rate is critical for surviving fluctuating environments in nature. But the enrichment for cancer-causing genes in the orthologous human subnetwork highlights the importance of this decision in disease biology, and it suggests that stress signaling in yeast may serve as a model for cancer signaling in humans. It is notable that orthologs of three key regulators in our network—Hog1, Cdc14, and CK2—have all been implicated in regulating the mammalian tumor suppressor p53 (Meek et al, 1990; Bulavin et al, 1999; Li et al, 2000), which controls the growth/survival/apoptosis decision in human cells and is mutated in many human cancers (Carvajal & Manfredi, 2013). These results underscore the importance of the growth/survival decision and hint that the yeast subnetwork could be used to implicate as-yet-unidentified human disease genes. An exciting area of future study will be to distinguish signaling dynamics and condition-specific versus common aspects of the signaling, with an eye toward their role in disease biology.

## Materials and Methods

### Growth conditions

All strains were of the BY4741 background, primarily from the deletion collection (Winzeler et al, 1999) (Thermo Scientific, Waltham, MA), except for cdc14-3 and its isogenic wild-type (kindly provided by Miller et al (2009)). BY4741 ckb1Δckb2Δ was kindly provided by Bergkessel et al (2011). Knockout strains were verified by diagnostic PCR to ensure correct integration of the drug cassette and to confirm absence of the deleted gene. Unless otherwise noted, cells were grown to log phase in batch YPD cultures at 30°C for at least seven generations before addition of a final concentration of 0.7 M NaCl, after which cells were grown for 30 min. cdc14-3 and isogenic wild-type cells were grown at 25°C, shifted to the non-permissive temperature of 35°C for 90 min (or 120 min for experiments from Fig 6G), and then treated with a final concentration of 0.7 M NaCl at 35°C for an additional 30 min before sample collection. Relative physiological changes were compared to the time point collected immediately before addition of NaCl (i.e., 35°C for 90 min without NaCl).

### Microarray analysis

Cell collection, RNA preparation, cDNA synthesis and labeling, array hybridization, and normalization were performed as previously described in Berry & Gasch (2008) and Lee et al (2011), using cyanine dyes (Flownamics, Madison WI) and Superscript III (Life Technologies, Carlsbad, CA). Samples were hybridized to whole-genome tiled DNA microarrays (Roche Nimblegen, Madison, WI), comparing cDNA from the salt-treated sample to cDNA generated from the unstressed culture. Dye orientation was performed on

select samples to assess dye-specific biases; dye orientation for paired mutant–wild-type samples was maintained for statistical analysis to avoid dye-specific effects. Comparison of unstressed strains was done as previously described Lee *et al* (2011) by retrieving and comparing single-channel data from mutant and wild-type arrays. Array data are available through the NIH GEO accession #GSE60613 and Supplementary Dataset S1.

Genes whose expression was altered in wild-type cells responding to NaCl were identified based on five biological replicates, using the Bioconductor package limma (Smyth, 2004) and Q-value (Storey *et al*, 2005) to assess the false discovery rate (FDR) and taking $q < 0.05$ as significant. This analysis identified 5,056 genes with a significant change in expression in response to NaCl. Genes with a defect in NaCl-responsive expression in mutants shown in Fig 1 were assessed in biological triplicate (for *hog1∆, mck1∆, pde2∆, msn2∆,* and *rim101∆* strains) or duplicate (all other strains). Expression defects were identified using contrast matrices to wild-type expression in limma for triplicated samples, with $q \leq 0.025$ taken as significant. For duplicated samples, expression defects were identified if both mutant replicates were outside the wild-type mean +2 standard deviations (95[th] confidence level), based on five replicates of the wild-type samples. Identified targets (summarized in Table 1) are available in Supplementary Dataset S2. Data for the *dot6∆tod6∆* mutant were taken from Lee *et al* (2011).

Validation experiments were performed on 10 deletion mutants responding to NaCl. For samples done in duplicate, significant expression changes were identified using limma with $q < 0.05$ taken as significant. Expression defects from singleton experiments were identified based on a 1.5-fold difference in expression in that mutant versus the paired wild-type sample. Identified expression defects are summarized in Table 1 and Supplementary Dataset S2. Expression in unstressed mutant cells was also assessed by comparing the mutant response to unstressed wild-type cells as described above. Unless otherwise noted, we detected few expression differences in unstressed cells. For Figs 6–8 where average expression values are plotted, data for some paired mutant–wild-type experiments (namely *cdc14-3, dot6∆tod6∆,* and *ck2* mutants) were scaled such that their paired wild-type data matched the plotted wild-type data taken from other experiments, in order to accurately represent those mutant defects by accounting for day-to-day variation of paired samples.

Expression analysis for Fig 8 was done in strains generously provided by JL Corden (West & Corden, 1995). Cells lacking endogenous *RPB1* carried a plasmid expressing *RPB1* with 14 wild-type CTD repeats (YSPTSPS) or a plasmid expressing chimeric *RPB1* genes: the Rpb1-CTD was composed of five repeats of S5A (YSP-TAPS) followed by seven wild-type sequenced repeats in the so-called S5A mutant, or 8 S2A repeats (YAPTSPS) followed by seven wild-type sequenced repeats in the S2A mutant. There was no difference in salt-responsive gene expression for control plasmids with 14 versus 21 wild-type repeats (not shown). Expression was measured as described above, before and at 30 min after treatment with 0.7 M NaCl. There were few expression differences in the strains before stress (see Fig 8B).

### Phospho-proteomic analysis

BY4741 was grown as described above, except that samples were taken before and at 5 min and 15 min after NaCl addition. Cells

were lysed by three passages through the French press at 4°C in 3 ml of lysis buffer consisting of 50 mM Tris pH 8, 4 M urea, 75 mM NaCl, 1 mM DTT, complete Mini EDTA-free Protease Inhibitor (Roche Diagnostics, Indianapolis, IN), and phosSTOP phosphatase inhibitor (Roche Diagnostics). The lysate was centrifuged at 14,000 $g$ for 10 min and the protein concentration determined by a bicinchoninic acid assay. Cysteine residues were reduced and alkylated by incubating lysate with 5 mM DTT for 45 min at 37°C followed by incubation in 15 mM IAA for 45 min at room temperature in the dark. After adding an additional aliquot of DTT to cap the alkylation reaction, the urea concentration was diluted to a final concentration of 1 M with 50 mM Tris and 1 mM $CaCl_2$. Proteins from each time point were digested overnight (37°C, pH 8) with trypsin (Promega, Madison, WI) at an enzyme:substrate ratio of 1:50. TFA was added to a final concentration of 0.5% to quench each digest, and the resulting peptides were desalted via solid phase extraction on a 50 mg $tC_{18}$ SepPak cartridge (Waters, Milford, MA) and the eluant lyophilized.

The desalted peptides from each time point were each labeled with a different tandem mass tag (TMT) isobaric label (Thermo-Pierce, Rockford, IL) according to the manufacturer's instructions. The differentially labeled TMT samples were pooled in equal volumes and dried-down. Labeled peptides were fractionated by strong cation exchange (SCX) on a polysulfoethyl A column (9.4 mm × 200 mm; PolyLC) with mobile phases A: 5 mM $KH_2PO_4$ pH 2.65 and 30% acetonitrile; B: 5 mM $KH_2PO_4$ pH 2.65, 350 mM KCl, and 30% acetonitrile; C: 5 mM $KH_2PO_4$ pH 6.5 and 500 mM KCl; and D: water. The gradient was generated by a Surveyor LC quaternary pump (Thermo Scientific, Waltham, MA) at 3 ml/min flow rate. Peptides were eluted over the following gradient and detected via a PDA detector (Thermo Scientific): 0–2 min, 100% A; 2–5 min, 0–10% B; 5–41 min 10–100% B; 41–48 min 100% B; followed by washes with C and D prior to re-equilibration with mobile phase A. Fifteen fractions were collected, lyophilized, and desalted. A small portion, 5%, of each was retained for unmodified protein analysis and the remaining material used for phosphopeptide enrichment.

Each fraction was enriched for phosphopeptides using immobilized metal ion affinity chromatography (IMAC). Magnetic beads (Qiagen, Valencia, CA) were washed three times with water, incubated with 40 mM EDTA (pH 7.5) for 30 min, and washed with water again. The beads were then incubated with 100 mM $FeCl_3$ for 30 min and washed four times with 80% acetonitrile and 0.1% TFA. Peptides from each fraction were resuspended in 1 ml of 80% acetonitrile and 0.1% TFA and incubated with the beads for 30 min. Unbound peptides were removed from the beads by washing four times with 80% acetonitrile and 0.1% TFA. Phosphopeptides were eluted using 1:1 acetonitrile:5% $NH_4OH$ in water, immediately acidified with 4% formic acid, and lyophilized.

Phosphopeptide-enriched and protein fractions were resuspended in 0.2% formic acid and analyzed by reverse-phase liquid chromatography on a nanoAcquity LC (Waters) coupled to an ETD-enabled LTQ Orbitrap Velos (Thermo Scientific). Samples were first loaded onto a 10 cm, 75 μm i.d. precolumn packed with 5 μm C18 particles (Bruker-Michrom, Fremont, CA) in 98% A (0.2% formic acid in water), 2% B (0.2% formic acid in acetonitrile) and then separated across a 25 cm, 50 μm i.d. analytical column packed with 5 μm C18 particles (Bruker-Michrom) using the following gradient: 0–3 min,

2–5% B; 3–123 min, 5–35% B; 123–133 min, 35–70% B; 133–138 min, 70% B; 138–165 min, 2% B. Phosphopeptide and protein fractions were each analyzed in duplicate. Methods to acquire mass spectra started with one MS1 survey scan ($R = 30,000$, 300–1,500 Th) followed by data-dependent MS2 fragmentation and analysis ($R = 15,000$) of the ten most intense precursors. The exclusion duration was 60 s for −0.55 Th to +2.55 Th of the sampled precursor. Ions with an unassigned charge state or a single charge were excluded. The QuantMode instrument control method was employed to reduce reporter ion interference caused by co-isolation of multiple precursors (Wenger *et al*, 2011a).

Spectral reduction was performed using DTA Generator. Generated text files were searched for fully tryptic peptides with up to three missed cleavages against a UniProt target-decoy database populated with yeast plus isoforms (downloaded 29 July 2011) using the Open Mass Spectrometry Search Algorithm (OMSSA) (Geer *et al*, 2004). Carbamidomethylation of cysteine (+57.021464), TMT 6-plex on lysine (+229.162932), and TMT 6-plex on peptide N-terminus (+229.162932) were searched as fixed modifications for all samples. Phosphopeptide-enriched fractions were additionally searched for variable phosphorylation modifications. Search results were filtered to 1% FDR at the unique peptide level and identified peptides quantified within the COMPASS software suite (Wenger *et al*, 2011b). Peptides were grouped into proteins according to previously reported rules (Nesvizhskii & Aebersold, 2005). Protein quantification was performed by summing all reporter ion intensities within each channel for each non-phosphorylated peptide mapping uniquely to that protein group.

Phosphorylation events were localized to specific residues using probabilistic methods (Phanstiel *et al*, 2011). Localized phosphorylated peptides were grouped together by identical modification sites, and their reporter ion intensities were summed. For simplicity, phosphorylation isoforms are referred to as phospho-sites. The average of two technical replicates was taken per time point, and phospho-sites with at least twofold change in recovery were taken as significant for downstream analysis. Average fold-changes of phospho-sites are available in Supplementary Dataset S3.

## Immunoprecipitation analysis

BY4741-*cka2Δ* and *cka1Δ* cells were transformed with empty vector or plasmids encoding GAL-inducible Cka2-GST or Cka1-GST (Zhu *et al*, 2001; Sopko *et al*, 2006) (Thermo Scientific), respectively. Cells were grown in YP-2% galactose medium in log phase to 0.6–0.8 $OD_{600}$, subjected to osmotic stress (0.7 N NaCl) for the indicated length of time, and lysed by bead-beating on ice. Cell lysates were incubated with glutathione Sepharose beads (GE Healthcare) at 4°C overnight in 1× PBS buffer with 1 mM DTT, 0.1% NP-40, 10% glycerol and protease inhibitors (Millipore, Billerica, MA). Proteins were eluted with 1× SB buffer and resolved by SDS–PAGE and detected by immunoblotting. Antibodies used were goat polyclonal anti-Hog1 (Santa Cruz Biotech, Dallas, TX), rabbit polyclonal anti-phospho-p38 MAPK (Cell Signaling), mouse monoclonal anti-actin (Pierce Biotech), goat polyclonal anti-Cdc14 (Santa Cruz Biotech) and goat polyclonal anti-GST (Abcam, Cambridge, MA). All blots shown in the manuscript are representatives of at least biological duplicates.

## Microscopy

Harvested cells were fixed with 4% final concentration of formaldehyde for 15 min, and GFP was visualized on a Leica DM LB2 microscope with standard GFP filters. DNA was detected via cell staining with 1 µg/ml DAPI for 5 min. Viability of *cdc14-3* cells was measured with Live-Dead staining (Life Technologies), which showed that NaCl-treated *cdc14-3* maintained viability close to WT cells for over 30 min after treatment with 0.7 M NaCl (not shown). Nuclear Hog1 was scored by visual inspection by comparing GFP signal to DAPI signal, in at least 75–100 cells per sample.

## *In vitro* CTD phosphorylation

Cells expressing C-terminally TAP-tagged proteins (Ghaemmaghami *et al*, 2003) (Thermo) were exposed to NaCl for the denoted times, snap-frozen, and then cryo-lysed with a Retsch Mixer Mill MM 400 as described in Churchman and Weissman (2011). Ground yeast was added to TAP Buffer A, and TAP-tagged kinase was purified as described in Puig *et al* (2001) and Liu *et al* (2004), with minor modifications. Kinases were eluted overnight at 4°C in 25 µl TAP Buffer A with 1 mM DTT and 10 U AcTEV (Invitrogen).

Peptide substrate GST-CTD14 (fourteen repeats of YSPTSPS fused to GST) was purified essentially as described in Patturajan *et al* (1998). Before elution, glutathione Sepharose beads were resuspended in 1 ml FastAP buffer (10 mM Tris–HCl pH 8.0, 5 mM $MgCl_2$, 100 mM KCl, 0.02% Triton X-100, 100 µg/ml BSA) and incubated with 100 U FastAP Thermosensitive Alkaline Phosphatase (Thermo Scientific) for 1 h at 37°C to remove any phosphates placed by the bacteria. Beads were washed, and GST-CTD14 was eluted. Any remaining alkaline phosphatase was heat-inactivated at 75°C for 5 min. The concentration of GST-CTD14 was determined via Bradford assays.

*In vitro* kinase assays were performed in at least biological duplicate using 5 µl of tandem affinity purified (TAP) kinase and 3 µM GST-CTD14 in 30 µl Buffer D as described in Ansari *et al* (2005), with minor modifications. For Hog1 inhibition assays, the kinase was pre-incubated with the inhibitor 4-(4-fluorophenyl)-2-(4-methylsulfinylphenyl)-5-(4-pyridyl)imidazole (Cell Signaling Technology) for 10 min prior to the reaction. Reactions were performed at 30°C for 2 h and resolved via SDS–PAGE and Western analysis using antibodies targeting CTD-Ser2P (Bethyl Laboratories), CTD-Ser5P (clone 3E8, gift from Dirk Eick), or the TAP tag (Thermo Scientific). Quantitation was performed using ImageJ. All images are representative of several biological replicates. The plot in Fig 5D shows background-subtracted levels of Ser2P and Ser5P normalized to Hog1-TAP abundance in each lane, then referred to levels seen in unstressed cells to calculate fold-change in phosphorylation.

## Analysis of novel predicted salt-response regulators

Fourteen predicted regulators not previously known to respond to NaCl were chosen for validation analysis. NaCl-responsive gene expression was measured in ten mutants, focusing on kinases and phosphatases not known to respond to NaCl and two RBPs (Scd6 and Arf3), as described above. Data for Rpd3, Bem1, Gal11, and Tpk2 were taken from previous studies probing the osmotic response (Alejandro-Osorio *et al*, 2009; Gitter *et al*, 2013), taking

$q < 0.05$ from limma $q$-value analysis as significant. Mutants were considered to have a defect in NaCl-dependent expression if there were at least fifty affected genes. Overlap between measured and predicted target genes was based on the hypergeometric test, scoring the probability of getting the number of observations or more compared to random expectation from the 3,330 genes used for IP input. Genes affected in unstressed cells were also identified (see above) and compared to predicted genes in cases where the NaCl-measured targets did not significantly overlap with predicted targets.

We also assessed the connections predicted between the interrogated regulators and other nodes predicted to lie in their paths. From the nodes predicted to lie in each regulator's path (based on the consensus-paths network), we identified those with known downstream targets (e.g., TF and RPBs) or targets measured in this study. We then scored the enrichment of each predicted node's known targets within the measured targets of the interrogated regulator, taking $P < 1e-6$ from the hypergeometric test as significant. Because the test lacks statistical power for large gene groups, we scored enrichment against the total list of measured targets as well as induced and repressed targets with defective or amplified expression changes considered separately. The results (Supplementary Table S2) indicate a lower bound of supported in-path nodes, since the hypergeometric test has lower statistical power for small gene groups (including known targets of several regulators), and targets of several in-path nodes were marginally enriched ($1e-5 < P < 0.01$) among measured targets of interrogated regulators but did not meet our stringent threshold. It is also possible that regulators that serve redundant roles are difficult to score with our assay, since single-gene knockouts may not identify all of the downstream targets.

**Chromatin immunoprecipitation (ChIP)**

ChIP was done similarly to as described in Tietjen *et al* (2010), on cells before and 20 min after treatment with 0.7 M NaCl. Rpb3 was immunoprecipitated using anti-Rpb3 antibody W0012 (Neoclone, Madison, WI) in strain Z26 carrying 'wild-type' or 'S5A' *RPB1* gene expressed on a CEN plasmid (West & Corden, 1995), described above. Chromatin was sonicated on a Misonix 4000 machine (Qsonica, Newtown, CT), input and immunoprecipitated material were amplified using ligation-mediated PCR as previously described Tietjen *et al* (2010) and hybridized to tiled Nimblegen arrays designed against the yeast genome (Lee *et al*, 2011). Data were normalized as in Tietjen *et al* (2010), except without the baseline adjustment procedure. All two-color arrays from two biological replicates were quantile-normalized together before further analysis. This procedure did not change any of the trends reported in the manuscript but helped to adjust the baseline across biological replicates done on different days. ChIP-chip data are available in the NIH GEO database under accession # GSE60613.

**Ortholog analysis**

To assess the relationship of the yeast consensus network ensemble to human diseases, we analyzed the orthologous set of human genes. We used the stringent RSD method of ortholog assignment (Wall *et al*, 2003), using a BLAST Evalue cut-off of 1e-5 and requiring fewer than 20% gapped positions in the global alignment. The

method identified 2,381 yeast-human orthologs; we focused on the 1,619 of these genes that are reviewed in humans. We compared these genes to those annotated in the COSMIC v67 (Forbes *et al*, 2011) and OMIM (Hamosh *et al*, 2005) databases. We also analyzed orthologous mouse proteins using the phenology.org database (Woods *et al*, 2013).

**Network Inference Methods**

*Background network for IP method*
To construct the background network, we identified a variety of binary interactions that are relevant to intracellular signaling and gene expression regulation. The background network, gathered from numerous public databases, represents interactions between pairs of proteins (Ptacek *et al*, 2005; Pu *et al*, 2009; Fasolo *et al*, 2011; Sharifpoor *et al*, 2011; Heavner *et al*, 2012), including kinase–substrate interactions, as well as protein–DNA interactions (Guelzim *et al*, 2002; MacIsaac *et al*, 2006; Everett *et al*, 2009; Ni *et al*, 2009; Abdulrehman *et al*, 2011; Venters *et al*, 2011; Huebert *et al*, 2012) and protein–RNA interactions (Hogan *et al*, 2008; Scherrer *et al*, 2010). After manual inspection of the background network neighborhoods of the interrogated mutants, we added a set of 17 missing interactions between the mutants and nearby regulators based on known interactions in the literature.

While the types of biological interactions in the background network are rich and diverse, we use a simplified representation as input to the computational method (illustrated in Figs 2 and 3A). The background network is represented as a graph, in which nodes represent genes and gene products, and edges represent interactions. A gene may be represented as two separate nodes in the background network: one representing the protein, and, for targets, one representing the DNA or mRNA. Each interaction may have a direction: for example, transcriptional regulatory interactions are directed, but most protein–protein interactions are not. The provenance of the background network and the interactions themselves are provided in Supplementary Information 1.2.3, Supplementary Table S2, and Supplementary Dataset S6.

*IP method input data and candidate paths*
The primary goal of the IP approach is to provide explanations for the salt-specific transcriptomic changes measured for this article. We also use two additional sources of salt-specific experimental data. From these data, we generate directed, acyclic candidate paths that serve as input to the IP (Fig 3B):

**Source–target pairs and paths, source–source paths** From the transcriptomic data measured in each of the original signaling mutants, we identified the set of downstream genes with dysregulated salt-responsive expression. We then extracted what we refer to as *source–target pairs*, each consisting of a single *source* protein and a *target* gene that was dysregulated in the source mutant under salt stress. Next, for each source, we used the hypergeometric test to identify candidate transcription factors (TFs) and RNA-binding proteins (RBPs) whose known binding targets (promoters or transcripts, respectively) are significantly enriched with the genes represented by the source's targets ($P < 0.05$). We also include TFs that are known to bind any number of targets under osmotic stress (Ni *et al*, 2009; Huebert *et al*, 2012). Candidate source–target paths

were enumerated to connect signaling mutants to their gene targets via candidate TFs/RBPs with up to three intermediate proteins between the source and TF/RBP (for a total of five interactions). Candidate path enumeration for each source was performed in an iterative deepening procedure, which was stopped at the path length at which at least 50% of candidate TFs/RBPs were reached.

**Fitness-contribution hits and hit–source paths** Previously, we identified yeast mutants that conferred a defect in acquired stress resistance after salt pretreatment (Berry *et al*, 2011). We refer to the gene products represented by these mutants as *fitness-contribution hits* because of the mutation's negative effect on yeast fitness under salt stress. Candidate hit–source paths and source–source paths were generated by finding short paths (including at most one intermediate protein) between these hits and the source proteins, and between pairs of source proteins. These paths are useful for interpreting the fitness-contribution hits in terms of connections to known regulators.

**Phospho-proteomic hits** We use this name to refer to the proteins that showed differential phosphorylation under salt stress.

**Receptor–source paths** Our method can take advantage of domain knowledge about the salt stress response in order to provide a scaffold for the inferred subnetwork. Here, we wanted to capture the most upstream stress sensors that may otherwise be missed in connecting sources to their downstream targets. We identified well-known indirect relationships between two transmembrane receptors, Sln1 and Sho1, and one of the sources, Hog1 (Saito & Tatebayashi, 2004). We enumerated candidate receptor–source paths (up to four intermediates) from Sln1 to Hog1 and Sho1 to Hog1 and provided them as input to the IP method.

Further details on the data and the generation of candidate paths are available in Supplementary Information 1.2.4. To measure the contribution of each input data set, we ran computational experiments in which each component was held aside. We also tested the effect of varying the length of the candidate paths. The results of these experiments are available in Supplementary Information Section 2.3 and 2.4.

*IP notation and variables*

The salt-specific signaling subnetwork is inferred by solving an integer linear program (IP, for short). We encode the relevance of each node, edge (physical interaction), and candidate path, and the direction of each edge, as binary variables. We characterize possible subnetworks using a set of linear constraints over those binary variables. Subnetwork inference is performed by choosing a union of relevant, directed paths that together satisfy our constraints and optimize a series of successively applied objective functions.

The values of some variables were determined by data provided as input to the inference process (for example, directions of directed edges), while others are inferred by solving the IP.

**Notation (summarized in Supplementary Table S3)** The input to the method is represented as a graph of nodes $\mathcal{N}$, edges $\mathcal{E}$, and candidate paths $\mathcal{P}$. A node represents either a protein or a target gene/mRNA. Protein nodes may belong to one or more of the following subsets: sources $\mathcal{N}^S$, fitness-contribution hits $\mathcal{N}^F$,

phospho-proteomic hits $\mathcal{N}^P$, and known membrane receptors $\mathcal{N}^R$. The set $\mathcal{N}^T$ describes targets, and for a given source node $n$, $\mathcal{N}^T(n)$, is the set of its targets.

The set of edges is $\mathcal{E} = (\mathcal{E}^D \cup \mathcal{E}^U)$, where $\mathcal{E}^D$ is the set of directed edges and $\mathcal{E}^U$ is the set of undirected edges. We denote an edge $e$ between nodes $n_i$ and $n_j$ as $e = (n_i, n_j)$. $\mathcal{N}(e)$ refers to the nodes connected by a particular edge $e$, and $\mathcal{E}(n)$ refers to the edges that touch a particular node $n$.

We consider four subsets of candidate paths $\mathcal{P}$: source–target paths between sources and their targets $\mathcal{P}^{ST}$, hit–source paths between fitness-contribution hits and sources $\mathcal{P}^{FS}$, source–source paths $\mathcal{P}^{SS}$, and receptor–source paths $\mathcal{P}^{RS}$ that connect known receptor proteins to sources. (Phospho-proteomic hits and additional fitness-contribution hits may appear in any of these paths.) To refer to the paths between a specific source $s$ and target $t$, we use the notation $\mathcal{P}^{ST}(s, t)$. We use the same notation to refer to other kinds of paths with specific endpoints: $\mathcal{P}^{FS}(f, s)$ $\mathcal{P}^{SS}(s_i, s_j)$, $\mathcal{P}^{RS}(r, s)$.

Each path $p$ specifies a direction for each of its undirected edges $e$, which is denoted as $dir(p, e)$. $\mathcal{E}(p)$ and $\mathcal{N}(p)$ refer to the edges and nodes in a particular path $p$.

**Variables (summarized in Supplementary Table S4)** The predicted relevance of a path $p$ is represented with the variable $\sigma_p$ which takes the value 1 if the path is included in the inferred subnetwork and 0 if it is not. As many as two variables describe each edge. The predicted relevance of an edge $e$ is represented with the variable $x_e$, which takes the value 1 if the edge is in at least one relevant path. For undirected edges in the background network, the variable $d_e$ represents the inferred direction of the edge. Each node $n$ has one variable: $y_n$, representing whether or not the node is present in any relevant paths. Finally, for all pairs of sources $(n_i, n_j)$, and also for all pairs consisting of one source and one fitness-contribution hit, the variable $c_{ij}$ represents whether or not the relevant subnetwork provides a directed path between the two nodes in the pair.

*IP constraints*

The following linear constraints define a subnetwork that, at minimum, provides consistently directed paths between source–target pairs and receptor–source pairs. Additional constraints are used to count up the number of connected fitness-contribution hit–source pairs and source–source pairs. These counts are optimized during the optimization procedure.

**Provide at least one path between each source–target pair** Each source must be connected to each of its targets by at least one relevant path. The following constraint requires that, for each source $s$, for each of its targets $t$, at least one source–target path $p$ in $\mathcal{P}^{ST}(s, t)$ from $s$ to $t$ must have $\sigma_p = 1$.

$$\sum_{\text{source–target paths } p \text{ in } \mathcal{P}^{ST}(s,t)} \sigma_p \geq 1 \qquad \begin{array}{l} \text{for all sources } s \text{ in } \mathcal{N}^S, \\ \text{targets } t \text{ in } \mathcal{N}^T(s) \end{array} \qquad (1)$$

**Provide at least one path between each receptor–source pair** We must provide at least one path showing the indirect relationship between an upstream receptor and a source. Similar to the previous constraint, this one requires that for each receptor $r$ and each of its

downstream sources $s$, there must be at least one receptor–source path $p$ in $\mathcal{P}^{RS}$ $(r, s)$ for which $\sigma_p = 1$.

$$\sum_{\text{receptor–source paths } p \text{ in } \mathcal{P}^{RS}(r,s)} \sigma_p \geq 1 \quad \begin{array}{l}\text{for all receptors r in } \mathcal{N}^R, \\ \text{source s in } \mathcal{N}^S (r)\end{array} \quad (2)$$

**Record whether or not there is a path between each fitness-contribution hit–source pair and source–source pair** Rather than require that each of these pairs is connected, we use the optimization procedure to maximize the total count of connected pairs. We use the following constraints to count up the number of connected pairs.

If there is a path between a fitness-contribution hit $f$ and a source $s$, set the variable $c_{fs} = 1$. Otherwise, set $c_{fs} = 0$:

$$\sum_{\text{hit–source paths } p \text{ in } (\mathcal{P}^{FS}(f,s) \cup \mathcal{P}^{FS}(s,f))} \sigma_p - c_{fs} \geq 0 \quad \begin{array}{l}\text{for all fitness-contribution} \\ \text{hits } f \text{ in } \mathcal{N}^F, \text{ sources } s \text{ in } \mathcal{N}^S\end{array}$$
$$(3)$$

$$c_{fs} - \sigma_p \geq 0 \quad \begin{array}{l}\text{for all fitness-contribution hits } f \text{ in } \mathcal{N}^F, \\ \text{sources s in } \mathcal{N}^S, \text{ hit–source paths } p \text{ in} \\ \mathcal{P}^{FS}(f,s) \cup \mathcal{P}^{FS}(s,f)\end{array} \quad (4)$$

Similarly, if source $s_i$ is connected to source $s_j$, we set $c_{ij} = 1$. Otherwise, set $c_{ij} = 0$.

$$\sum_{\text{source–source paths } p \text{ in } (\mathcal{P}^{SS}(s_i,s_j) \cup \mathcal{P}^{SS}(s_j,s_i))} \sigma_p - c_{ij} \geq 0 \quad \begin{array}{l}\text{for all pairs of sources} \\ (s_i, s_j) \text{ in } \mathcal{N}^S \times \mathcal{N}^S\end{array}$$
$$(5)$$

$$c_{ij} - \sigma_p \geq 0 \quad \begin{array}{l}\text{for all pairs of sources } (s_i, s_j) \text{ in } \mathcal{N}^S \times \mathcal{N}^S, \\ \text{source–source paths } p \text{ in } \mathcal{P}^{SS}(s_i, s_j) \cup \mathcal{P}^{SS}(s_j, s_i)\end{array} \quad (6)$$

**All edges in a relevant path are relevant** For an edge $e$ to be relevant (that is, have $x_e = 1$), there must be at least one relevant path that contains it (that is, a path $p$ for which $\sigma_p = 1$). Similarly, a relevant path $p$ must contain all relevant edges $e$. The set $\mathcal{P}(e)$ refers to the paths that contain edge $e$.

$$\sum_{\text{paths } p \text{ in } \mathcal{P}(e)} \sigma_p - x_e \geq 0 \quad \text{for all edges } e \text{ in } \mathcal{E} \quad (7)$$

$$x_e - \sigma_p \geq 0 \quad \text{for all paths } p \text{ in } \mathcal{P}, \text{ edges } e \text{ in } \mathcal{E}(p) \quad (8)$$

**All nodes in a relevant edge are relevant** A node $n$ is relevant if it is connected to a relevant edge $e$ (where $x_e = 1$). Each node $n$ for a relevant edge $e$ must be relevant ($y_n = 1$).

$$\sum_{\text{edges } e \text{ in } \mathcal{E}(n)} x_e - y_n \geq 0 \quad \text{for al l nodes } n \text{ in } \mathcal{N} \quad (9)$$

$$y_n - x_e \geq 0 \quad \text{for all edges } e \text{ in } \mathcal{E}, \text{ nodes } n \text{ in } \mathcal{N}(e) \quad (10)$$

**All paths must be uniquely directed** For a relevant path $p$, all undirected edges $e$ in that path ($e$ in $\mathcal{E}(p) \cap \mathcal{E}^U$) must be uniquely oriented so that the path proceeds only in one direction. This required direction for each edge is determined when the candidate path is generated, and is given by $dir(p, e)$. (For source–target paths, the required direction allows the path to proceed from the source to

the target.) The term including $I(\cdot)$, the indicator function, returns 1 if an edge's inferred direction corresponds to the direction that the path requires for it.

$$I(d_e = dir(p,e)) - \sigma_p \geq 0 \quad \begin{array}{l}\text{for all paths } p \text{ in } \mathcal{P}, \\ \text{undirected edges } e \text{ in } \mathcal{E}(p) \cap \mathcal{E}^U\end{array} \quad (11)$$

*Solving the IP to find an ensemble of subnetworks*
An optimal inferred subnetwork satisfies two goals: maximizing the inclusion of salt-response-relevant proteins that are supported by experimental evidence, and minimizing the number of additional nodes that are necessary for connecting each source to each target. To achieve this, we apply four successive objective functions. To accompany the following description, a diagram of the process is depicted in Supplementary Fig S8.

To model and solve the IP, we used the GAMS modeling system v. 23.9.3 and the ILOG CPLEX solver v. 12.4.0.1. Both are commercial packages for which an academic license available at a reduced cost. We provide our GAMS code in Supplementary Dataset S7.

**Step 1: Maximize connections between hits and sources** This involves solving the IP to identify max_connections, the maximum number of connections possible between pairs of sources, and between pairs of fitness-contribution hits and sources. The purpose of this step is to reveal proximal connections between salt-responsive proteins, whether or not they occur between sources and targets. In this constraint, the set $(\mathcal{N}^S \times \mathcal{N}^S) \cup (\mathcal{N}^F \times \mathcal{N}^S)$ gives all source–source pairs and fitness-contribution hit–source pairs, and the sum counts up the number of pairs that are connected by relevant paths.

$$\text{max\_connections} = \max \sum_{\text{hit–source pairs}(n_i,n_j) \text{ in } (\mathcal{N}^S \times \mathcal{N}^S) \cup (\mathcal{N}^F \times \mathcal{N}^S)} c_{ij} \quad (12)$$

After optimizing this criterion, we add a new constraint to the IP:

$$\sum_{\text{hit–source pairs}(n_i,n_j) \text{ in } (\mathcal{N}^S \times \mathcal{N}^S) \cup (\mathcal{N}^F \times \mathcal{N}^S)} c_{ij} = \text{max\_connections} \quad (13)$$

**Step 2: Maximize inclusion of fitness and phospho hits** Next, we solve the IP to identify max_hits, the maximum number of fitness-contribution hits and phospho-proteomic hits that can be included in the relevant subnetwork. This step prioritizes the use of nodes with experimental evidence of being relevant to the salt stress response.

$$\text{max\_hits} = \max \sum_{\text{nodes } n \text{ in } (\mathcal{N}^F \cup \mathcal{N}^P)} y_n \quad (14)$$

After identifying the maximal number of hits that can be included in the subnetwork, we add a new constraint to the IP:

$$\sum_{\text{nodes } n \text{ in } (\mathcal{N}^F \cup \mathcal{N}^P)} y_n = \text{max\_hits} \quad (15)$$

**Step 3: Minimize total nodes and find multiple solutions** Now, we solve the IP with a new objective function, which minimizes the number of nodes required to satisfy all of the constraints. The resulting

subnetwork will include only those nodes that are required to explain the experimental data.

$$\min_{\text{nodes } n \text{ in } \mathcal{N}} \sum y_n \tag{16}$$

At this point, we find an ensemble of solutions to the IP, where each solution identifies a minimum set of nodes (while still satisfying all other constraints). The CPLEX solver allows for the identification of multiple solutions. First, the CPLEX solver uses a branch-and-cut algorithm to find one optimal solution; this algorithm entails maintaining a tree of linear relaxations of the IP. Next, the solver proceeds down previously rejected branches of the tree to identify additional optimal solutions with different variable settings. For our experiments, we identified 10,000 solutions.

**Step 4: Maximize the number of paths in each solution** After predicting the relevant nodes in the previous step, we would like to see all possible relevant connections between them, to aid in their interpretation. For each of the solutions identified in the previous step, we solve the IP again to maximize the number of relevant directed paths between the nodes included in the solution. This step does not change the node content of each solution, but instead reveals all possible directed paths that connect the node set chosen in the previous step.

For each solution:

First, we introduce constraints to fix each value of $y_n$ to its value from the previous solution, $\widehat{y_n}$:

$$y_n = \widehat{y_n} \quad \text{for all nodes } n \tag{17}$$

Next, we maximize the number of relevant paths:

$$\max_{\text{paths } p \text{ in } \mathcal{P}} \sum \sigma_p \tag{18}$$

At this point, we assemble the solutions into an ensemble of inferred subnetworks. Using the ensemble, we assign a confidence value for a prediction based on the number of solutions in the ensemble that support the prediction. We performed several experiments to assess the effect of each component of our four-part objective function, as well as their ordering. These can be found in Supplementary Information Sections 2.3 and 2.5.

*Precision–recall analysis*

To assess the predictive accuracy of the ensemble (as shown in Fig 4B), we curated a list of true positives and a list of likely negative proteins. True positives were defined as genes previously identified in the Hog network based on literature curation (de Nadal & Posas, 2010; Tiger *et al*, 2012), genes with 'osmotic' or 'osmolarity' in their *Saccharomyces* Genome Database (SGD) (Cherry *et al*, 2012) annotations, and genes with 'stress regulator' in their SGD annotations, if they were also linked to the osmotic response in at least one publication. In all, this identified 112 true positives. Likely negatives were taken as genes with no evidence for nuclear localization and whose GO compartment annotation was 'mitochondrion', 'mitochondrial envelope', 'peroxisome', 'vacuole',

'Golgi', and/or 'endoplasmic reticulum'. Proteins annotated in SGD as 'metabolic enzymes' were also added to this list of likely negatives. From this list, we removed 32 well-known signaling proteins, many of which were already on the true positive list; in all, this left 1,865 likely negative proteins for the network assessment. Among these test cases, the background network contained 108 positives and 1,512 likely negatives. In order to separate out the effect of the experimental hits on predictive accuracy, we omitted all hits from the test cases, leaving 70 true positives and 1,416 likely negatives. For each test case (true positive or likely negative), we measured the inferred subnetwork ensemble's confidence that it is relevant to the salt response. This is calculated as the fraction of the 10,000 solutions in which the test case appears as a protein node in the subnetwork.

We compared our ensemble's precision–recall curve to two baselines, which we refer to as the *candidate* baseline (Fig 4B, green) and *permuted* baseline (Fig 4B, yellow). For the candidate baseline, we computed the precision and recall of the test cases using the complete set of protein nodes present in candidate paths. For the permuted baseline, we compared the inferred ensemble's accuracy to that of a set of 1,000 ensembles inferred using permuted experimental data. For each of 1,000 permutations, we randomly drew a set of sources, proteins with fitness defects, and proteins with phospho-changes from the background network, equal in number and degree distribution to the true experimental data. To generate receptor–source pairs, we randomly drew two proteins from the background network and paired each with a randomly chosen source. To generate permuted source–target pairs, for each source, we randomly drew an equal number of targets from the entire background network. We inferred an ensemble of 1,000 solutions for each permutation and measured the confidence of each test case as the average confidence over all 1,000 ensembles.

*Ranking of putative ESR bifurcation points*

We constructed the salt-relevant ESR consensus subnetwork shown in Fig 7A and C as follows. First, we gathered three clusters of genes defined by (Gasch *et al*, 2000) based on expression profiles under multiple stress conditions: iESR (induced ESR) and two rESR (repressed ESR) subclusters, RiBi and RP. Using the protein–nucleic acid interactions from the background network, we identified potential transcriptional regulators of the three ESR gene clusters. These were TFs and RBPs whose targets were enriched for a cluster (determined by hypergeometric test, using a threshold of FDR = 0.1, calculated by the Benjamini–Hochberg procedure). For iESR targets, we identified 25 total potential TFs/RBPs, of which 22 are TFs and three are RBPs. We found 16 TFs and 10 RBPs for the combined rESR clusters.

Next, we extracted the consensus source–target paths (having confidence ≥ 75%) that end in an interaction between an ESR-relevant TF/RBP and ESR-relevant target gene (of the same cluster). For each protein node in each ESR-relevant consensus path, we assigned a label based on the ESR cluster(s) represented by the downstream ESR-relevant TF/RBPs. These labels were used to perform the coloring in Fig 7. Finally, we removed the targets that were not a member of any ESR cluster.

Using the ESR consensus paths, we identified candidate bifurcation points, defined as nodes that are upstream of both rESR

                                                                              

and iESR targets (yellow and orange nodes in Fig 7), according to how well their outgoing paths show a distinct division between the induced and repressed clusters. To rank the candidates, we defined a bifurcation score, $B(n)$, that is related to the concept of information gain ratio (Quinlan, 1986). $B(n)$ is calculated as follows. An illustration of this process is provided in Supplementary Fig S9.

First, we define the *count C (T)*, which counts the number of bits required to represent the cluster membership of all of the targets in a set $T$. Considering the clusters $c \in \{$iESR, rESR$\}$, let $T^c(n)$ be the set of targets downstream of $n$ that belong to the ESR cluster $c$.

$$C(T(n)) = - \sum_{\text{clusters } c \text{ in \{iESR,rESR\}}} |T^c(n)| \log_2 \frac{|T^c(n)|}{|T(n)|} \quad (19)$$

An ideal bifurcation point would have a high $C(T(n))$ compared to the paths that emanate from it. To perform this comparison, we next calculate $C(\cdot)$ for each of the paths downstream from $n$. If the subnetwork were a tree, $n$'s targets would simply be partitioned by $n$'s children. However, since the paths leading out from $n$'s children may converge on the same targets, we instead partition $T(n)$ into disjoint subsets of targets, each of which is reachable via a unique combination of $n$'s children. We refer to $n$'s outgoing partitions as $P_1(n)\ldots P_m(n)$.

After having calculated $C(P_i(n))$ for each partition, we then calculate the *information gain*, $I(n)$, which measures the number of bits that are saved by partitioning the targets downstream of $n$:

$$I(n) = C(T(n)) - \sum_{i=1}^{m} C(P_i(n)) \quad (20)$$

Finally, to calculate the bifurcation score $B(n)$, we normalize $I(n)$ by the *split information $S(n)$*, which measures the number of bits required to describe the *partition* assignment of one of $n$'s targets. $I(n)$ is strongly biased toward nodes whose outgoing partitions split each target each into its own partition. The normalized score $B(n)$ prioritizes nodes that have a small number of (relatively) cleanly split outgoing paths and many downstream targets.

$$S(n) = - \sum_{i=0}^{m} \frac{|P_{i(n)}|}{|T(n)|} \log_2 \frac{|P_{i(n)}|}{|T(n)|} \quad (21)$$

$$B(n) = \frac{I(n)}{S(n)} \quad (22)$$

The complete ranking of the 92 candidate bifurcation points (yellow and orange nodes in Fig 7) according to $B(n)$ is available in Supplementary Dataset S9.

**Data availability**

Microarray and ChIP-chip data are available in the NIH GEO database under accession #GSE60613. Proteomic data are available in the Chorus mass spectrometry repository under accession #YeastSaltStress.

**Supplementary information** for this article is available online: http://msb.embopress.org

## Acknowledgements

Thanks to S. Topper and A. Peterson for experimental and computational assistance, and M. Weinreich, C. Guthrie, F. Posas, A. Kumar, D. Eick, J.L. Corden for sharing strains and reagents, and M. Ferris for computational assistance and resources. Data are available in Supplemental Materials and NIH GEO accession GSE60613. This work was funded by NIH R01 GM083989 to A.P.G., R01 GM080148 to J.J.C., NSF MCB0747197 to A.Z.A, NSF IIS-1218880 and NIH UL1 RR025011 to M.C., and NIH Training Grants in Genomic Sciences (T32HG002760 for ALM), Computation and Informatics in Biology and Medicine (5T15LM007359 for DC), and Chemistry–Biology Interface Training Program (T32 GM008505 for CMN). CMN was also supported by an NSF Graduate Research Fellowship.

## Author contributions

DC, MC, DBB, and APG conceived of the project; DC, MC, and APG designed computational approach which was implemented by DC; AEM, MVL, JJC oversaw and conducted proteomic work; YHH, CMN, and AZA collaborated on CTD studies; YHH, DBB, MM, JH, and JW conducted expression analysis; YHH performed all other molecular experiments, APG oversaw the project, and all authors contributed to writing the manuscript.

## Conflict of interest

The authors declare that they have no conflict of interest.

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
