## [Review Process File · Molecular Systems Biology]

High cross-pathway connectivity in the yeast stress-activated signaling network

Deborah Chasman, Yi-Hsuan Ho, David B. Berry, Corey M. Nemece, Matthew E. MacGilvray, James Hose, Anna E. Merrill, M. Violet Lee, Jessica L. Will, Joshua J. Coon, Aseem Z. Ansari, Mark Craven and Audrey P. Gasch

Corresponding author: Audrey P. Gasch, University of Wisconsin-Madison

Review timeline:

Submission date:	19 January 2014
Editorial Decision:	27 February 2014
Revision received:	07 April 2014
Editorial Decision:	25 April 2014
Revision received:	24 August 2014
Editorial Decision:	30 September 2014
Revision received:	03 October 2014
Accepted:	15 October 2014

Editor: Maria Polychronidou

Transaction Report:

1st Editorial Decision

27 February 2014

Thank you again for submitting your work to Molecular Systems Biology. We have now heard back from the three referees who agreed to evaluate your manuscript. Overall, the reviewers acknowledge that the presented analysis is comprehensive and results in potentially interesting findings. However, as you will see from the reports below, they raise concerns on your work, which should be convincingly addressed in a revision of the manuscript.

Without repeating all the points listed below, some of the more fundamental issues refer to the need to better substantiate the integration of microarray and phosphoproteomic measurements in the model and to justify several of the modeling decisions. Moreover, the referees mention the lack of a comparison of the present approach to existing methods and datasets. As mentioned by referee #3, some follow-up experimentation to further support the proposed role of Cdc14 and the phosphorylation of the RNA Polymerase II CTD in the stress response would strengthen the overall biological insights.

If you feel you can satisfactorily deal with these points and those listed by the referees, you may wish to submit a revised version of your manuscript. Please attach a covering letter giving details of the way in which you have handled each of the points raised by the referees. A revised manuscript will be once again subject to review and you probably understand that we can give you no guarantee at this stage that the eventual outcome will be favorable.

Reviewer #1:

Chasman et al. have applied an integer programming approach to a network of mixed interactions from existing resources integrated with new molecular profiles in WT and mutant backgrounds. The results of this analysis identified existing and new factors of signaling networks important for the signaling osmotic stress response. This ambitious project, diverse in both data and methods, offers a number of novel predictions some of which were validated by follow-up analyses. In particular, the findings related to the role of Cdc14p and PolIII in this response are interesting and the expression and proteomic data sets would be of general interest.

Major comments:

1. Clarification of aims and focus: A challenge when reading was to keep the aim of the study in focus. The presentation and emphasis of the IP methodology often seemed to overshadow the experimental approach, novel findings and lead the reader to (likely mistakenly) interpret the central focus as the IP method. This creates a problem due to the lack of comparison to existing methods for pathway inference. At the minimum, and in all cases, other types of methods should be mentioned in the introduction (e.g. Bayesian inference, Factor graphs and methods inspired by electronic circuits). In particular, Gat-Viks et al. 2005 applied factor graphs to infer signaling and regulation in hyperosmotic stress in budding yeast. See also Yeang, Ideker, and Jaakkola 2004, Schadt et al. Nat. Gen. 2005, Kulp and Jagalur BMC Genomics 2006, Lee, Pe'er et al. PNAS 2006; Tu et al. Bioinformatics 2006, Perez-Enciso et al. BMC Genomics 2007.
2. An overview of experimental approach, profiling methods, selected mutant strains and data sets used in the interference is missing. The number and types of experiments was too difficult to find in methods section and should be stated earlier. A figure or table before current Fig 2 would help to resolve this.
3. An overview figure or table of the experimental results is also missing. Plenty of gene sets are described in the text but the reader is left to put the pieces together themselves to see the overall impact of the stress in the various mutants (with the exception of Fig 7B which is likely too late in the presentation to serve this purpose).
4. Influence of highly connected nodes (hubs) on the IP inference results is unclear. Were protein kinases enriched in the findings for the reason of their biological function/relevance or because they have a tendency to have a high node degree in the background network?

Minor comments:

1. "mitotic exists" -> "mitotic exit"
2. "pathways were intimately connected" is probably too anthropomorphic in this context
3. Mix of standard *S.cerevisiae* gene names and generic gene names should be avoided, e.g. "CK2, Yak1, Bck1"
4. "confirm the CTD" -> "confirm the Rb1 CTD"
5. Figure 3A is not informative
6. It is unclear how 5056 genes were found differentially expressed in the salt stress. Was this only for WT or over all mutants studied? I don't believe previously studies implicated >80% of genes in osmotic stress.
7. Interaction data sets used to build the background network should be briefly listed in the main text.

Reviewer #2:

The present study is an interesting attempt to integrate different data layers and prior knowledge of protein interactions and transcription regulation into a consistent, integrated network, predictive of the signaling mechanisms and their interconnectivity to integrate the NaCl stress response in yeast.

This work is very comprehensive, both in terms of multiple data layers to constrain the inferred network to realistic insight, as well as in terms of validation (experimental and literature-based). We also applaud that authors make code, data, and background and inferred network available.

For network inference, the authors used an elegant solution, consisting of a simple IP optimization procedure, which has been used before. What seems new in this contribution and an interesting contribution to the community is the assumption of 4 different paths to enable optimization using two different layers of data. The constraints in the multi-layered objective function are simple, and the computational method not tremendously novel, but is an important contribution in that the method allows the authors to effectively gain insight regarding connectivity between pathways and contribution of signal integration hubs.

Some major concerns are detailed below that we feel have to be addressed by the authors. These pertain to how the microarray and the phosphoproteomic measurements are included in their models as mediators and targets. In addition, the authors should properly characterize their approach and support the modeling decisions made throughout and discuss their method in context with other similar approaches, as commented below.

Major revisions:

(1) The authors have made a number of decisions regarding modeling, including:

- i) page 14, paragraph entitled "Source target pairs and paths, source-source paths". The authors identify candidate paths including at most 4 intermediate proteins. How does that affect the results?
- ii) page 14, paragraph entitled "fitness defect hits and hit-source paths". The authors identified candidate hit-source paths including at most one intermediate. How does this affect model predictions?
- iii) The authors use 4 objective functions sequentially, what is the reason for the specific order they used? Does the order affect model predictions? Could they have been lumped together into a single objective function?

The authors need to discuss whether these decisions affect their model and in what way. Is it possible the sequential optimization results in a local rather than global solution?

(2) Another question would be regarding the first constraint of page 15. The authors force their algorithm to fit all source target pairs. Given that gene expression data may be noisy do they have a mechanism to cope with this noise? This could have major impact on the results.

(3) Related with the previous question, assume a target gene downstream of a transcription factor that is active upon a perturbation. The algorithm will identify a source-target path from that perturbation through that transcription factor and to the target gene. And since the IP forces at least one relevant path between all source-target pairs, this path will be conserved and added to the consensus network. However, assuming that TF may have 200-300 more targets genes, all inactive apart from that one target gene, we believe that inferring this TF to be active because one of its target genes is active is likely incorrect. If that TF was active, then there should be more evidence in the gene expression data, not only one gene. Or differently put, the reviewer suggests that there should be a set percentage of target genes active for every TF to be included in the solution.

(4) Phosphosignals: The decision as to how to consider proteins with both increased and decreased p-sites could be more rigorous - why 2-fold change?. How do you handle when there are multiple p-sites with non-consistent fold changes? This would be very relevant to understand the role of said proteins in signal transduction and interconnectivity.

(5) Interconnectivity: It has been largely showed and reviewed that network architecture is dynamic and changes through time and space depending on the context. Here the authors exhaustively test different network mutants, but the context in terms of stress is kept constant. Could they discuss/quantify how their findings in terms of network connectivity would be affected by changing the environmental conditions, in particular different doses of NaCl stress?

(6) Availability of method: We applaud that the authors provide the code, data, background and inferred networks in cytoscape format. However, they could describe better how their tool could be used by others in different datasets. What are the requirements in terms of data? model identifiability? How scalable is the approach?

(7) The authors challenge their method via precision/recall analysis with different networks. While this is done in a question-driven manner (e.g. is the initial network sufficient to guarantee predictivity?), figure 3B is unclear and does not display well how this is performed. Could they label figure 3B more meaningfully, and could they add other measurements of predictivity?

(8) The authors generate 14 mutants lacking the predicted regulators. How were these chosen out of the 380 nodes?

Minor/text suggestions:

- (-) The statistical test used on the microarray dataset determines the targets to be used. In the results

main text, it is a bit unclear how this is done and difficult to read. The authors state "A third of the affected genes were dependent on {greater than or equal to}2 regulators, and there was significant overlap in several target-gene sets". Do they mean that one can see common patterns across rows, or did they actually quantify this? The word "significant" here may be misleading. Same applies for "genes dependent on the NaCl-activated Hog1 kinase, the PKA-inhibiting Pde2 phosphodiesterase, and nutrient-responsive Rim15 and Mck1 kinases not previously linked to NaCl stress all displayed statistically significant overlap." If there was indeed a test to quantify overlap, provide p-value.

(-) Regarding cross-talk, it is unclear what the authors conclude after observing that the inferred network contains the STE mating pathway "The inclusion of the mating pathway indicates that some connections in the consensus subnetwork represent signaling suppression that prevents crosstalk to other pathways.". Could they phrase this differently?

(-) The IP formulation is not written clearly. Please remove brackets from the constraints and write the formulation in the standard form:

Min/max THE OBJECTIVE FUNCTION

Subject to:

THE CONSTRAINTS in the form $A \cdot x \leq B$; i in ...; j in ...; k in ... etc

(-) Second constraint in page 18 has a special character typo in "max_connections"

(-) Please number all equations in the manuscript

Reviewer #3:

The manuscript by Chasman and collaborators addresses the integration of several sets of data to create a comprehensive regulatory network in yeast. The data collected includes transcriptome, phosphoproteomics as well as previous data on other aspects from cells that have been challenged with salt stress. From these sets of data and the inclusion of previous known players in this specific response the authors create a network. Several predictions are made based on the information from the network such as a new role for the protein phosphatase Cdc14, a known cell cycle regulator, the regulation of the CTD of the RNA polymerase and the connection between HOG and CK2 pathways (via Cdc14). In addition, the authors claim that the computational method serves to establish connections in human based in their predictions from yeast.

At this point, the network created to understand the response to salt stress is clearly insufficient based on the data included in it. The authors include an initial set of transcriptome profiles based on a very particular selection of mutants (based on a previous article from the same group), however, a number of transcriptome analysis have been made by a number of groups that could be included here. Also, the phosphoproteome data included in the data set is not contrasted with previous phosphoproteomic analysis made by different groups in response to salt stress, thus it is difficult to know how the data overlap with extensive phosphoproteomic analyses. Therefore, albeit the attempt to create a signaling network might be of interest, the lack of internal controls and complete data sets poses doubts on the final outcome of the analysis.

Furthermore, and even more problematic is the fact that there is a complete lack of solid experimental support for some of the predictions of the network analysis. The authors claim that Cdc14 phosphatase plays a key role in the response, however, there is no biological insight on how Cdc14 might be regulated by salt, what Cdc14 is doing to reduce transcription or whether this is just an indirect effect of an essential gene. Also, the data on phosphorylation of the RNA PolII CTD and the regulation of transcription is clearly insufficient. The fact that CTD phosphorylation is reduced in a hog1 mutant can simply reveal a decrease on overall transcription in response to salt that is not compensated in mutant cells. If more is to be claimed, then data supporting a direct phosphorylation of the CTD, the relevance of this phosphorylation in gene transcription and its regulation should be provided. The authors claim that there is a connection between the HOG and CK2 pathways. However, how this connection is established and the relevance of it is not shown here. It is reasonable to assume that not all the predictions from a network analysis are supported by experimental data; however, if the network can not be validated in some of its more important conclusions then it is difficult to believe. The claim that this is relevant in mammals without any

additional data is also very difficult to support.

1st Revision - authors' response

07 April 2014

Reviewer #1:

Chasman et al. have applied an integer programming approach to a network of mixed interactions from existing resources integrated with new molecular profiles in WT and mutant backgrounds. The results of this analysis identified existing and new factors of signaling networks important for the signaling osmotic stress response. This ambitious project, diverse in both data and methods, offers a number of novel predictions some of which were validated by follow-up analyses. In particular, the findings related to the role of Cdc14p and PolIII in this response are interesting and the expression and proteomic data sets would be of general interest.

First, we thank the reviewer for their positive assessment of the work overall.

Major comments:

1. Clarification of aims and focus: A challenge when reading was to keep the aim of the study in focus. The presentation and emphasis of the IP methodology often seemed to overshadow the experimental approach, novel findings and lead the reader to (likely mistakenly) interpret the central focus as the IP method. This creates a problem due to the lack of comparison to existing methods for pathway inference. At the minimum, and in all cases, other types of methods should be mentioned in the introduction (e.g. Bayesian inference, Factor graphs and methods inspired by electronic circuits). In particular, Gat-Viks et al. 2005 applied factor graphs to infer signaling and regulation in hyperosmotic stress in budding yeast. See also Yeang, Ideker, and Jaakkola 2004, Schadt et al. Nat. Gen. 2005, Kulp and Jagalur BMC Genomics 2006, Lee, Pe'er et al. PNAS 2006; Tu et al. Bioinformatics 2006, Perez-Enciso et al. BMC Genomics 2007.

We have edited and extended the Introduction to clarify the goals of this study and to better acknowledge related work (pages 2-3). We have included many of the citations suggested by the reviewer as well as other relevant works about inferring networks from diverse data sources, including genetic and environmental perturbation studies. We have also expanded the comparison to related methods in the Discussion.

2. An overview of experimental approach, profiling methods, selected mutant strains and data sets used in the interference is missing. The number and types of experiments was too difficult to find in methods section and should be stated earlier. A figure or table before current Fig 2 would help to resolve this.

To aid in the overview description of our methods and datasets, we added a new figure (new Figure 2) that describes the experimental approach and outlines the profiling methods and datasets. We also listed the number of gene targets identified for each of the 16 'source' regulators in Table 1A, which we moved from the supplement to the main text

3. An overview figure or table of the experimental results is also missing. Plenty of gene sets are described in the text but the reader is left to put the pieces together themselves to see the overall impact of the stress in the various mutants (with the exception of Fig 7B which is likely too late in the presentation to serve this purpose).

We have moved the previous Supplementary Table 1 into the main manuscript's Table 1. This table summarizes the mutants analyzed (both for the initial data generation and the validation analysis), the number of replicates, and the number of affected downstream genes. We also provide two figures on the supplement that summarize the overlap in gene targets (Supplemental Figure 2 and Supplemental Figure 4). We hope this helps to clarify the datasets, which are very complex and otherwise difficult to summarize visually.

4. Influence of highly connected nodes (hubs) on the IP inference results is unclear. Were protein

kinases enriched in the findings for the reason of their biological function/relevance or because they have a tendency to have a high node degree in the background network?

The reviewer raises an important point, which is that kinases may be enriched in the final subnetwork simply because they are ‘hubs’ of high degree in the background network. Indeed, there is also some enrichment for kinases in the permuted subnetwork analysis, which is a control for the type of background structure that the reviewer points out (Supplementary Table 5, ‘Permutations’ row). We have therefore reworded this section on Page 5 to state that enrichment for kinases may be due to their high connectivity in the background network.

Minor comments:

1. *"mitotic exists" -> "mitotic exit"*

This typo has been corrected, thank you for pointing it out.

2. *"pathways were intimately connected" is probably too anthropomorphic in this context*

We changed the text to state, “pathways were intricately connected”

3. *Mix of standard S. cerevisiae gene names and generic gene names should be avoided, e.g. "CK2, Yak1, Bck1"*

We have replaced “CK2” throughout the manuscript and supplement with “CK2 subunits ..” followed by the appropriate subunit names (except in cases in which we refer to the human CK2 kinase).

4. *"confirm the CTD" -> "confirm the Rrb1 CTD"*

The text has been updated to state “confirm the Rpb1 CTD”

5. *Figure 3A is not informative*

We realize that these subnetwork diagrams are often difficult to parse. However, we prefer to leave the subnetwork diagram as one of the figure panels, because it shows graphically the proportion of the network made up by kinases, transcription factors, and RNA binding proteins as well as the degree (represented by node size) and the proportion of the network represented by proteins with phospho-changes. Therefore, at this stage we have retained this panel as part of the figure.

6. *It is unclear how 5056 genes were found differentially expressed in the salt stress. Was this only for WT or over all mutants studied? I don't believe previously studies implicated >80% of genes in osmotic stress.*

We performed five replicates of WT responding to a fairly high dose of 0.7M NaCl, using tiled genomic DNA arrays and careful culture/sampling conditions, which together give extremely high statistical power when identifying differentially expressed genes. This number of differentially expressed genes in WT cells is consistent with other studies in our lab using these very sensitive methods and high number of replicates. Visual inspection of the clustered data is consistent with the false-discovery rate used as a cutoff.

7. *Interaction data sets used to build the background network should be briefly listed in the main text.*

We have added a sentence to page 4 and Methods citing the main databases from where the background data were taken.

Reviewer #2:

The present study is an interesting attempt to integrate different data layers and prior knowledge of protein interactions and transcription regulation into a consistent, integrated network, predictive of the signaling mechanisms and their interconnectivity to integrate the NaCl stress response in yeast.

This work is very comprehensive, both in terms of multiple data layers to constrain the inferred network to realistic insight, as well as in terms of validation (experimental and literature-based).

We also applaud that authors make code, data, and background and inferred network available.

For network inference, the authors used an elegant solution, consisting of a simple IP optimization procedure, which has been used before. What seems new in this contribution and an interesting contribution to the community is the assumption of 4 different paths to enable optimization using two different layers of data. The constraints in the multi-layered objective function are simple, and the computational method not tremendously novel, but is an important contribution in that the method allows the authors to effectively gain insight regarding connectivity between pathways and contribution of signal integration hubs.

We thank the reviewer of their positive assessment of the work.

Some major concerns are detailed below that we feel have to be addressed by the authors. These pertain to how the microarray and the phosphoproteomic measurements are included in their models as mediators and targets. In addition, the authors should properly characterize their approach and support the modeling decisions made throughout and discuss their method in context with other similar approaches, as commented below.

Major revisions:

(1) The authors have made a number of decisions regarding modeling, including:

i) page 14, paragraph entitled "Source target pairs and paths, source-source paths". The authors identify candidate paths including at most 4 intermediate proteins. How does that affect the results?

We have performed a few additional computational experiments varying the length of source-target paths. In the original manuscript (as described in Supplementary Information, Section 1.2.4, "Candidate-source-target paths"), we searched for paths using an iterative deepening search, stopping after five interactions or when at least 50% of a particular source's candidate TFs/RBPs were reached. In the additional experiments performed in response to the reviewer's question, we tried three variations on the path length, stopping the iterative deepening search at lengths three, four, and five. At a path length of five, our IP was unable to find a solution due to the large number of paths generated by two sources, Hog1 and Mck1. As a compromise, we truncated the paths for those sources to four, but searched for paths of length five for all other sources.

We evaluated the resulting subnetworks based on the same precision-recall, enrichment and stability analyses that we performed in the lesion experiments in Supplementary Information Section 2.2. The results of these analyses do not appear to be very sensitive to the path length. The precision-recall curves are visually very similar, and proportions of relevant proteins are unaffected. As path length increases, there is a slight increase in recall and the method's ability to account for more of the input data, a decrease in precision, a small but significant decrease in genetic interaction enrichment, and an increase in computational complexity.

We discuss the results of these experiments (as well as those prompted by the next reviewer comment) in a new Supplementary Information Section 2.5, "Testing variations on candidate path length". We show the precision-recall curves in Supplementary Figure 13A, stability analysis results in Supplementary Figure 13B, and enrichment comparisons in a new Supplementary Table 10.

ii) page 14, paragraph entitled "fitness defect hits and hit-source paths". The authors identified candidate hit-source paths including at most one intermediate. How does this affect model predictions?

Our lesion testing experiments partially addressed the question by assessing the removal of hit-source/source-source paths entirely (results reported in Supplementary Information 2.3, "Maximizing the number of connections between hits and sources"). We have now also performed an additional experiment to evaluate the effect of extending and reducing the length of these paths by one interaction. We compare both variations to the manuscript's choice of two interactions (one protein intermediate). Decreasing hit-source/source-source path length results in an increase in precision and genetic interactions, but no statistically significant change in the proportion of relevant proteins among high-confidence nodes. Increasing the path length results in a clear decrease in precision as well as a small but significant decrease in genetic interaction enrichment. Applying a limit to the length of these upstream paths appears to be useful. The stringency of the limit should be

chosen depending on how the inferred subnetwork is to be used to guide further experiments. By allowing one intermediate node to these paths, the method can offer connections for more fitness defect hits, while sustaining a small decrease in precision.

The experiment is discussed in Supplementary Information 2.5, “Hit-source and source-source paths”. Precision-recall curves are shown in Supplementary Figure 13C, stability analysis results in Supplementary Figure 13D, and the enrichment scores are shown in the new Supplementary Table 10.

iii) The authors use 4 objective functions sequentially, what is the reason for the specific order they used? Does the order affect model predictions? Could they have been lumped together into a single objective function?

The authors need to discuss whether these decisions affect their model and in what way. Is it possible the sequential optimization results in a local rather than global solution?

We have clarified our motivation for the multi-part objective function in the Methods subsection “Solving the IP to find an ensemble of subnetworks”. The high-level goal of the inference process is to maximally explain the experimental data while using a minimal subnetwork. Data are explained using both nodes and paths, and so for technical reasons we use four objective functions rather than two. It would certainly be possible to lump together all objective functions into a single function. However, doing so would require us to assign a weight to each component. By running the objective functions sequentially, we are able to prioritize the components without introducing other parameters. We prioritize the components according to the following narrative.

- 0) The primary goal of the inference process is to explain the source-target pairs using directed paths. Therefore, we enforce that all source-target pairs are connected by using a hard constraint to the IP. The IP also specifies that these paths must be directed.
- 1) A secondary goal is to explain the fitness defect hits and identify nearly-direct connections between them in the background network. We believe it is useful to see which hits are closely connected to many of our tested sources. We cannot require that all possible connections are made due to possible conflicts over edge directions. Therefore, we use the first objective function to maximize the number of these connections, subject to the requirement for directed paths that connect all source-target pairs.
- 2) We want the subnetwork to provide connections to proteins for which we have some evidence of relevance to the salt response. We use the second objective function component, “Maximize the number of fitness contribution and phospho-proteomic hits”, to make sure that the inferred subnetwork contains as many of these hits as possible.
- 3) The third component imposes a preference for a parsimonious subnetwork by minimizing nodes. In combination with the previous component, this function enforces a preference to include hits over proteins without any evidence of relevance.
- 4) Finally, the fourth component, “maximize the number of paths”, does not change the node composition of the inferred subnetworks. Instead, we use it to identify all possible directed paths between the nodes chosen by the fourth component.

The effect of each component on the inferred subnetwork is partially addressed by the lesion testing experiments presented in Supplementary Information 2.3. In one of these experiments, we assessed the effect of individually holding aside each component of the objective function. In this section, we empirically demonstrated the worth of each component, or argued for the value of the component because it enhances interpretability.

Also to further address the reviewer’s comment, we performed an additional experiment in which we re-ordered the “minimize” and “maximize” portions of the objective function sequence. These results are provided in Supplementary Information Section 2.4, Supplementary Figure 12 and Supplementary Table 9.

(2) Another question would be regarding the first constraint of page 15. The authors force their algorithm to fit all source target pairs. Given that gene expression data may be noisy do they have a

mechanism to cope with this noise? This could have major impact on the results.

With regard to the gene expression data, we used relatively stringent false-discovery rate cutoffs to identify downstream targets of specific signaling proteins. Therefore, the ‘noise’ in the expression data is likely to lead to false negatives (*i.e.* missed targets) rather than false positive associations.

Any false calls are likely to have only a minor influence on the results for two reasons: first, we do not force all targets to be connected to upstream signaling factors, and therefore the method can accommodate some errors in the individual source-target calls. But more importantly, source regulators are connected to downstream targets through TF and RBP regulators, only if the set of known TF/RBP targets is enriched in the measured targets of that source regulator. In other words, the group of TF/RBP targets must be affected in order to connect an upstream source with its downstream genes via a path through that TF/RBP. The text describing this in the manuscript on Page 4 has also been clarified to convey these points.

(3) Related with the previous question, assume a target gene downstream of a transcription factor that is active upon a perturbation. The algorithm will identify a source-target path from that perturbation through that transcription factor and to the target gene. And since the IP forces at least one relevant path between all source-target pairs, this path will be conserved and added to the consensus network. However, assuming that TF may have 200-300 more targets genes, all inactive apart from that one target gene, we believe that inferring this TF to be active because one of its target genes is active is likely incorrect. If that TF was active, then there should be more evidence in the gene expression data, not only one gene. Or differently put, the reviewer suggests that there should be a set percentage of target genes active for every TF to be included in the solution.

The reviewer raises a very important point, and in fact this is what we already did in the subnetwork inference. We addressed this potential problem by filtering the TFs/RBPs prior to inference. For each source, we use the hypergeometric test to identify a set of candidate TFs/RBPs whose binding targets (as a group) are enriched with the source’s downstream targets (as a group) with $p < 0.05$. As an exception, we include any TF that differentially binds targets during osmotic stress (Huebert et al., 2012, Ni et al., 2009). This procedure is explained in detail in Supplementary Information 1.2.4, “Candidate source-target paths,” paragraph 3. We have now adjusted the main manuscript text to emphasize this filtering step. We also clarified this step in the newly added Figure 2, which provides an overview of the data handling before IP inference.

(4) Phosphosignals: The decision as to how to consider proteins with both increased and decreased p-sites could be more rigorous - why 2-fold change?. How do you handle when there are multiple p-sites with non-consistent fold changes? This would be very relevant to understand the role of said proteins in signal transduction and interconnectivity.

Interpreting the meaning of phosphorylation changes is unfortunately in its infancy – increases in protein phosphorylation can be activating or inhibiting, as can decreased phosphorylation. As the reviewer points out, there are often multiple phospho-sites per protein. Therefore, at this state it is impossible for us to interpret the meaning of the phospho-changes. We primarily considered phospho-changes to implicate proteins that may have *some* functional change in response to NaCl and thus to give preference to subnetwork paths that include proteins with phospho-changes. We have clarified the text to say that we consider proteins with a 2X change in phosphorylation at one or more residues. Due to limits in the quantification of protein phosphorylation, we applied a conservative two-fold cutoff to avoid false positive calls.

(5) Interconnectivity: It has been largely showed and reviewed that network architecture is dynamic and changes through time and space depending on the context. Here the authors exhaustively test different network mutants, but the context in terms of stress is kept constant. Could they discuss/quantify how their findings in terms of network connectivity would be affected by changing the environmental conditions, in particular different doses of NaCl stress?

We have added a short discussion to this effect in the final paragraph of the Discussion.

(6) Availability of method: We applaud that the authors provide the code, data, background and inferred networks in cytoscape format. However, they could describe better how their tool could be

used by others in different datasets. What are the requirements in terms of data? model identifiability? How scalable is the approach?

Accessibility. The GAMS code we provide with the manuscript was generated from our background network and dataset. In order to run our inference procedure on a different dataset, one would need to generate candidate paths specific to that data set and recreate the GAMS code in “nacl_data_sets.gms”. We have updated the readme.txt file included with the code to clarify the scope of the provided code.

Scalability. Many factors affect the scalability of the method. We have not performed experiments to quantify scalability, but can summarize some important factors here:

Factors affecting the scalability of candidate path generation:

- Size of background network
- Length of requested paths
- Number of candidate TFs/RBPs for each source
- Number of and network connectivity of source-target pairs

Factors affecting the scalability of inference:

- Number of candidate paths
- Number of undirected edges in need of orientation
- Number of IP solutions requested
- Number of threads available to CPLEX for parallel execution

In our experiments, we allowed CPLEX to use from 24 to 78 threads to solve the first three objective functions (including the accumulation of 10,000 minimal node sets). This took approximately 10 minutes. For each node set, we solved again to maximize the number of directed paths; each of these solutions took approximately 1 minute. The runtime of this final optimization was not improved by adding more threads, so we used 4. This final step can also easily be split into several batches and run in parallel. We split the 10,000 solutions into 10 batches, and found that it took approximately 13 hours to solve each batch of 1,000 solutions.

Identifiability. We cannot make any claims about the identifiability of our models given the noise in our experimental measurements, the lack of a complete background network, and the existence of multiple, optimal subnetworks. However, Supplementary Section 2.2 does consider the related notion of stability. The results in this section show that measures such as node relevance, edge relevance, etc. tend to be highly stable.

(7) The authors challenge their method via precision/recall analysis with different networks. While this is done in a question-driven manner (e.g. is the initial network sufficient to guarantee predictivity?), figure 3B is unclear and does not display well how this is performed. Could they label figure 3B more meaningfully, and could they add other measurements of predictivity?

We have clarified the explanation of the precision-recall analysis on page 5. In addition to the precision-recall analysis, we compared the consensus subnetwork to the candidate, background, and permuted networks by testing their enrichment with stress regulators. These comparisons are explained in the main text on page 5, Supplementary Information 2.1, and Supplementary Table 5.

(8) The authors generate 14 mutants lacking the predicted regulators. How were these chosen out of the 380 nodes?

We clarified the text on page 5 as follows, “To test some of the novel predictions of the inference, we analyzed osmo-dependent transcriptome changes in 14 mutants lacking predicted regulators (Fig S4 and Supplemental Information). We gave preference to kinases and included proteins activated by other stresses but not implicated in the NaCl response (Yak1, Bck1, Pho85), proteins with little known function in stress-dependent gene regulation (Kin2, Nnk1, Scd6, Arf3), and several others that are either poorly characterized or for which other datasets exist to test downstream effects (CK2 subunits Cka2, Cka1, Ckb1/2; Tpk2, and Bem1).”

Minor/text suggestions:

(-) The statistical test used on the microarray dataset determines the targets to be used. In the

results main text, it is a bit unclear how this is done and difficult to read. The authors state "A third of the affected genes were dependent on {greater than or equal to}2 regulators, and there was significant overlap in several target-gene sets". Do they mean that one can see common patterns across rows, or did they actually quantify this? The word "significant" here may be misleading. Same applies for "genes dependent on the NaCl-activated Hog1 kinase, the PKA-inhibiting Pde2 phosphodiesterase, and nutrient-responsive Rim15 and Mck1 kinases not previously linked to NaCl stress all displayed statistically significant overlap." If there was indeed a test to quantify overlap, provide p-value.

For this analysis, overlap in downstream targets was scored using the hypergeometric test with Bonferroni-corrected p-values. We have added a statement to this effect in the text and the legend to Figure 1, which represents the $-\log(p\text{-value})$ of each pairwise overlap in targets with colored edges.

(-) Regarding cross-talk, it is unclear what the authors conclude after observing that the inferred network contains the STE mating pathway "The inclusion of the mating pathway indicates that some connections in the consensus subnetwork represent signaling suppression that prevents crosstalk to other pathways.". Could they phrase this differently?

We clarified the text slightly to indicate that, because this is a known suppressive interaction between the Hog1 pathway and the Ste pathway (which is not activated by NaCl unless *HOG1* is deleted), the inclusion of the Ste pathway raises the possibility that other pathways/connections in the subnetwork could represent suppression of crosstalk. This sets the stage for our later results, showing that some of the Cdc14 connections represent Cdc14-dependent suppression of signaling to the cell cycle network.

(-) The IP formulation is not written clearly. Please remove brackets from the constraints and write the formulation in the standard form:

Min/max THE OBJECTIVE FUNCTION

Subject to:

THE CONSTRAINTS in the form $Ax \geq B$; i in ...; j in ...; k in ... etc

In writing our IP, we followed the previous examples provided by Ourfali et al. (2007), Gitter et al. (2011), and Silverbush et al. (2011), but introduced brackets and logical quantifiers. Similar to those publications, we introduced the constraints first (which describe directed paths), and the objective function second (which describe how optimal subnetworks are chosen from possible feasible solutions to the constraints). Like Ourfali et al. and Silverbush et al., we provided a textual explanation for each constraint. In response to the reviewer's suggestion, we have updated the presentation of the constraints. We a) minimized the use of brackets, b) replaced the logical quantifier characters with full words ("for i in ..."), and c) adjusted the spacing of the constraints to make them easier to read. This presentation is also consistent with that of Silverbush et al.

(-) Second constraint in page 18 has a special character typo in "max_connections"
Thanks; we have fixed this typo.

(-) Please number all equations in the manuscript
We now number equations in the manuscript.

Reviewer #3:

The manuscript by Chasman and collaborators addresses the integration of several sets of data to create a comprehensive regulatory network in yeast. The data collected includes transcriptome, phosphoproteomics as well as previous data on other aspects from cells that have been challenged with salt stress. From these sets of data and the inclusion of previous known players in this specific response the authors create a network. Several predictions are made based on the information from the network such a new role for the protein phosphatase Cdc14, a known cell cycle regulator, the regulation of the CTD of the RNA polymerase and the connection between HOG and CK2 pathways

(via Cdc14). In addition, the authors claim that the computational method serves to establish connections in human based in their predictions from yeast.

At this point, the network created to understand the response to salt stress is clearly insufficient based on the data included in it. The authors include an initial set of transcriptome profiles based on a very particular selection of mutants (based on a previous article from the same group), however, a number of transcriptome analysis have been made by a number of groups that could be included here.

This reviewer is concerned that by not broadly including other datasets done by other labs that our network cannot be complete.

We have added a statement to the Introduction that clarifies that, while there have been many insightful studies analyzing yeast response to osmotic shock, it is important here that we use only data generated under the same conditions (e.g. same strain, culturing conditions, osmolyte, salt dose, etc). It is now well known (in part by prior work in our lab), that the genes and even signaling proteins important for the stress response are highly context dependent – using different strains, and even different base medium or stress dose, can have a significant effect on the downstream response, including on the regulatory proteins that are activated. Therefore, we feel that including other datasets from other labs and experimental conditions is not appropriate, as nice as those datasets are – it will interfere with the inference for the reasons outlined above.

However, we have now more clearly outlined our rationale in the text on Page 3 and cited some of the other prior studies that have been done before this work.

Also, the phosphoproteome data included in the data set is not contrasted with previous phosphoproteomic analysis made by different groups in response to salt stress, thus it is difficult to know how the data overlap with extensive phosphoproteomic analyses.

We have performed an analysis between our yeast phospho-proteome response to 0.7M NaCl to a prior study by Soufi et al. that followed the yeast phospho-proteome upon challenge with 0.4 M NaCl. We added a short section to the supplement (legend to Supplementary Figure 3) that describes that, not surprisingly given the challenges in phospho-proteome analysis, there is very low overlap between the two studies. This could be due to substantial differences in the culturing conditions and stress dose, or it could simply be due to the low overlap in peptide sampling, a well known limitation of phospho-proteomic mass spec analysis that produces low overlap even for technical replicates of the same lysate digestion.

Therefore, albeit the attempt to create a signaling network might be of interest, the lack of internal controls and complete data sets poses doubts on the final outcome of the analysis.

We hope we have convinced this reviewer why it is not appropriate to include other datasets in our network inference. We point out that we went to great lengths to validate the network, both computationally through precision-recall analysis and functional enrichments and experimentally through a large number of microarray validations and extensive downstream validation through targeted molecular analysis (see more below).

Furthermore, and even more problematic is the fact that there is a complete lack of solid experimental support for some of the predictions of the network analysis. The authors claim that Cdc14 phosphatase plays a key role in the response, however, there is no biological insight on how Cdc14 might be regulated by salt, what Cdc14 is doing to reduce transcription or whether this is just an indirect effect of an essential gene.

We respectively disagree with the reviewer on this point. Our goal here was not to work out every detail of Cdc14's role in the NaCl response, but rather to validate new *connections* and *predictions* made by the network, which will seed future detailed study. In this regard, we remain awed by how many of the subnetwork predictions were validated. The network predicted a connection between Cdc14 and CK2 subunit Cka2, and we found a NaCl-enhanced interaction (notably, in an otherwise wild-type cell demonstrating that the involvement of Cdc14 cannot be explained by a generic defect in a mutant). The network predicted overlap between Cdc14 targets and Cka2 targets, and we

validated this through transcriptome analysis in the corresponding mutants. The network predicted that Cdc14 regulates Hog1 via nuclear transporters, and we validated that Hog1 nuclear localization upon NaCl treatment is defective in a way that cannot be explained by a generic defect in the mutant or a response specific to G2/M-arrested cells.

As requested by this reviewer, we have also added one more set of experiments to further showcase the network predictions. We initially reported that the *cdc14-3* mutant has an aberrant, strong activation of G1 and S-phase genes upon NaCl treatment, even though the cells are completely arrested in M phase. These results suggest that Cdc14 normally suppresses signaling crosstalk to the cell-cycle network. To better understand this response, we again turned to the subnetwork, which predicts that Cdc14 is connected to these downstream genes via Snf1, a kinase well known to be activated by NaCl but which has also recently been implicated in proper timing of cell-cycle entry under standard conditions. We measured NaCl-responsive gene expression in *snf1Δ* cells as well as a *cdc14-3 snf1Δ* double mutant. Remarkably, *SNF1* deletion in the *cdc14-3* mutant partially abrogates the cell cycle expression response (even though cells remain arrested in M-phase, just like the *cdc14-3* mutant). These results strongly suggest that Cdc14 somehow suppresses Snf1's effect on the cell cycle network (Cdc14 and Snf1 have a measured interaction in the background network), allowing Snf1 to regulate other downstream genes related to carbon metabolism. Certainly, we do not fully understand all the details of this crosstalk – but again that was not our goal here. Our goal was to show that the subnetwork makes many new predictions that are accurate and that the subnetwork opens a new view of the stress physiology, one that will seed future study in areas that we did not know about before.

Also, the data on phosphorylation of the RNA Pol II CTD and the regulation of transcription is clearly insufficient. The fact that CTD phosphorylation is reduced in a hog1 mutant can simply reveal a decrease on overall transcription in response to salt that is not compensated in mutant cells. If more is to be claimed, then data supporting a direct phosphorylation of the CTD, the relevance of this phosphorylation in gene transcription and its regulation should be provided. The authors claim that there is a connection between the HOG and CK2 pathways. However, how this connection is established and the relevance of it is not shown here. It is reasonable to assume that not all the predictions from a network analysis are supported by experimental data; however, if the network can not be validated in some of its more important conclusions then it is difficult to believe. The claim that this is relevant in mammals without any additional data is also very difficult to support.

With regard to Rpb1 CTD, we have a separate manuscript in which we provide detailed follow-up of the predictions made here. Again, one of our goals here was simply to show that the subnetwork allows us to make new predictions about biology to foster future hypothesis-driven research. We believe that our experiments clearly validate the subnetwork predictions: we confirm the well-known interaction between Hog1 and Rpb1, we show that purified Hog1 can phosphorylate the Rpb1 CTD *in vitro*, we show that Hog1-dependent phosphorylation of the CTD is blocked by a Hog1-specific inhibitor added *in vitro*, and we show that the *hog1Δ* mutant has a defect in bulk CTD phosphorylation which, given the direct *in vitro* phosphorylation by Hog1 and the fact that Hog1 is known to travel with polymerase (Nadal-Ribelles *et al.* 2012 and Cook *et al.* 2012), is consistent with direct regulation by Hog1. We believe that these experiments provide a great showcase for the predictions of the subnetwork, which again was our primary goal with these follow-up experiments. The same can be said for our confirmation that Hog1 and Cka2 physically interact, analogous to what is already known in human cells (but has not been directly reported before in yeast).

The reviewer does make an important point, that there will certainly be some false-positive associations (including errors in predicted players and the edges between them). We have added a statement to this effect in the Discussion section on Page 11.

In addition to the revisions requested by the reviewers, we also made small additional changes:

- In discussing the computational methodology, we standardized our terminology throughout the document for the genes with fitness contributions. We now call them "fitness-contribution hits" (rather than "fitness-defect hits").
- We identified and corrected an error in which a set of results was reported in Supplementary Table

6 (now Supplementary Table 5). The submitted table contained results based on the consensus subnetwork derived from the ensemble of 1,000 solutions, which we used for the lesion testing and other computational experiments discussed in Supplementary Information Section 2. However, the intention was to show the results for the 10,000-solution consensus subnetwork used for all evaluations and validations in the main manuscript. At $p < 0.05$, all significant comparisons are the same between the two ensemble sizes, although the p -values are slightly different. We now provide the correct table.

2nd Editorial Decision

25 April 2014

Thank you again for submitting your revised work to Molecular Systems Biology. We have now heard back from the two referees who agreed to evaluate your manuscript. As you will see from the reports below, the referees still raise significant concerns regarding the conclusiveness of the study.

In particular, referee #3 is still not convinced that the experimental validation is sufficient to conclusively support the presented novel biological findings. Along the same lines, referee #2 recommends the inclusion of further validations. Since the presentation of novel biological insights is important given the scope of Molecular Systems Biology, we agree with the referees that inclusion of further data is required.

While in principle, Molecular Systems Biology only allows a single round of major revision, we would be willing to make an exception in this case and offer you the opportunity to submit another revision of this work. We would understand however if you would prefer to maintain the manuscript in its current form and submit it elsewhere.

If you feel you can provide additional experimental data better substantiating the main findings and you do wish to resubmit, we would ask you to attach a covering letter giving details of the way in which you have handled each of the points raised by the referees. A revised manuscript will be once again subject to review and you probably understand that we can give you no guarantee at this stage that the eventual outcome will be favorable.

Reviewer #2:

We thank the authors for the thorough revision of the manuscript to address our concerns, as well as those of the other reviewers.

We feel the authors have convincingly addressed our questions and, as far as we can judge, those of the other reviewers.

A final point: as mentioned by reviewer 3, there is no dedicated validation of the findings within this manuscript. Authors mention a separate manuscript about Rbp1 CTD. While we consider the content of the current manuscript already significant, inclusion of that (or another) validation would provide a much more complete and appealing publication.

Reviewer #3:

The authors have clarified some of the concerns raised by some of the reviewers. Still, the experimental data to support the validity of the study is rather limited.

The answer to this request was that this was not the goal of the study. I agree that the goal was to create a network, however, from the analysis the made a number of predictions that albeit, some data was collected to support them, still the results are not conclusive and additional interpretations can be done based on the results presented here.

Also, the authors decided not to include additional data on phosphoproteomics just because the overlap between their set and the published in Soufi et al, is rather limited. May be the best approach

was to include both in the analysis.

2nd Revision - authors' response

24 August 2014

Chasman & Ho *et al*, Response to Reviewers

Both Reviewer 1 and Reviewer 2 were very positive in their initial set of reviews; we addressed all of their initial comments in the previous set of revisions. In response to that revision, Reviewer 2 said that we had ‘convincingly addressed’ all of their comments and those of the other reviewers (Reviewer 1 did not re-review the work). Reviewer 2 said that they considered the last version of the manuscript an already significant contribution, but in light of Reviewer 3’s comments said that specific biological follow-up on one of the main points of the manuscript would provide a more compelling story.

Reviewer 3 stated that we had clarified some of their concerns but that more validation of the main findings made from the network analysis was required.

To address this remaining concern, we have added several new, substantial experiments testing one of the main predictions of our work – that direct modification of the RNA Pol II carboxyl-terminal domain (CTD) is a key regulatory point coordinating induction of stressdefense genes with repression of growth-related genes.

In the previous version of the manuscript, we showed that TAP-purified Hog1 can phosphorylate the Rpb1-CTD *in vitro*, in a manner dependent on prior salt treatment and inhibited by a Hog1- specific inhibitor added *in vitro*. We also showed that the *hog1* mutant shows a defect in saltresponsive Rpb1-CTD phosphorylation *in vivo*, compared to wild type cells. Later in the manuscript, we explored the network to investigate how signaling is bifurcated to coordinate induction of iESR genes with repression of rESR genes. We proposed that regulation of RNA Pol II is a key point in this decision.

In the new version of the manuscript, we have added substantial biological follow-up validating this model. First, we measured transcriptome changes in cells harboring mutant Rpb1-CTD proteins in which Serine 2 or Serine 5 is substituted with alanine (S2A and S5A mutants, respectively). Neither mutant had a major expression defect before stress; the S2A mutant had only minor differences in salt-responsive transcriptome changes. In contrast, the S5A mutant had a major defect in iESR induction and an even bigger defect in rESR repression, similar to the defects seen in the *hog1* mutant. These data are shown in the new Figure 8B.

To further test our model, we followed localization of Pol II subunit Rpb3 before and after stress in wild-type and CTD-S5A mutant cells, through CHIP-chip studies. The S5A mutant responding to salt stress had a reproducible defect in Rpb3 depletion over rESR genes and failed to recruit Rpb3 to iESR promoters or genes.

Together with our other data, these results validate our model: that direct phosphorylation of the Rpb1-CTD, in part by Hog1, is required to coordinate repression of rESR genes with induction of iESR genes, by triggering relocalization of RNA Pol II. We believe that this is an important result, not only because it provides new insight into Pol II and ESR regulation, but also because it highlights our ability to make fundamental new insights using our inferred signaling network.

To accommodate the new data in an already weighty manuscript, we streamlined the text to focus on the main points and reorganized the text slightly: data from the previous Fig 6 has been moved to Fig 5D and Fig 8A, and the new experiments have been added to new Fig 8B, Fig 8C, and Supplementary Fig 7A-C. Some details from the main body have been moved to the supplement (changes to the supplement are highlighted with colored text).

This manuscript provides substantial biological validation of the network and its predictions. In addition to the new experiments (Fig 8), our manuscript presents many experiments that characterize Cdc14’s central role in the network (including several experiments added in the last revision, Fig 6), validates the role of new regulators implicated by the network (with substantial discussion and seven biology figures on the supplement), and tests the predictions of upstream ESR regulators (Fig 7). We have highlighted text describing our extensive biological validations (including all the new experiments) in grey. We hope that the addition of new experiments described here has met the reviewer’s requirements. In the process of reworking the manuscript, we do believe that the story is more compelling and that the manuscript provides a more focused read.

Thank you again for submitting your work to Molecular Systems Biology. In this revised manuscript you have performed further experiments to examine the involvement of RNA PolII phosphorylation in ESR coordination. In summary, you now show that:

- TAP-purified Hog1 phosphorylates Rpb1-CTD Ser2 and Ser5 *in vitro*, only after treatment of the cells with NaCl (Fig. 5D).
- In Hog1 cells responding to NaCl, Ser5 and Ser2 Rpb1-CTD phosphorylation initially decreases (as in the wild type) but does not subsequently increase as would be expected (Fig. 8A).
- While the S2A non-phosphorylatable mutant shows only subtle defects in NaCl-dependent expression, the S5A mutant shows defects in iESR induction and rESR repression (Fig. 8B). (Gene expression profiling was performed using cells expressing chimeric CTD sequences with half mutant and half wild-type repeat sequences.)
- S5A mutant cells display a defect in Rpb3 recruitment to iESR genes and Rpb3 release from rESR genes (ChIP-Chip experiments, Fig. 8C).

We have now heard back from the two referees who agreed to evaluate your manuscript. As you will see below, reviewer #2 is satisfied with the additional experiments and thinks that they enhance the overall biological significance of the work. However, reviewer #3 mentions that: "the authors do not show that this Ser5 phosphorylation is mediated by Hog1 *in vivo* and that this is the mechanism for ESR regulation". We circulated the reports to all reviewers as part of our 'pre-decision cross-commenting' policy and we enquired whether they would recommend toning down statements such as "Hog1 directly phosphorylates the Rpb1-CTD in response to stress" and "direct modification of the Rpb1-CTD is required for coordinated regulation of iESR and rESR genes". Both referees agreed with our suggestion to avoid overstatements regarding the phosphorylation of Rpb1-CTD by Hog1. As such, we would like to ask you to modify the text accordingly, in a revision of this work.

On a more editorial level, we would like to ask you to include the following when you submit the revised version of your work:

- We would like to ask you to deposit the proteomics and phosphoproteomics data in one of the major public databases and to provide the dataset identifiers in the "Data Availability" section of your manuscript.

Reviewer #2:

We are happy with the revised version of the manuscript. In our opinion, this revision has addressed our concerns as well as those of the other reviewer. The revisions have strengthened the quality of an already high-quality manuscript.

We thank the authors for extensive validation studies and appreciate the streamlining of the manuscript in the updated version. Through these validation studies using both transcriptomics and chip-chip newly acquired datasets, the authors confirm the role of Rpb1 CTD, which upon phosphorylation is shown here to coordinate the repression of rESR genes with induction of iESR genes. Altogether, we believe this work presents an elegant and efficient approach for context-dependent subnetwork inference, and we anticipate that readers of MSB will be interested in more widely using these methods on future datasets - hence we can recommend it for publication.

Reviewer #3:

This is a re-revised manuscript by Chasman & Ho and collaborators. A major criticism to the initial study was the lack of solid experimental support for some of the predictions of the network analysis. After an initial round of revision, the authors added some experimental data to support their claims. However, the data provided was not sufficient to support the initial claims since only showed

indirect evidence of the suggested connections between the yeast Hog1 MAPK and downstream actors. In this second submission the authors provide an additional set of experiments trying to demonstrate only one of the predictions from the analysis that is the phosphorylation of the Pol II CTD by the MAPK.

From the data presented the authors conclude that "that direct phosphorylation of the Rpb1-CTD, in part by Hog1, is required to coordinate repression of rESR genes with induction of iESR genes, by triggering relocalization of RNA Pol II. We believe that this is an important result, not only because it provides new insight into Pol II and ESR regulation, but also because it highlights our ability to make fundamental new insights using our inferred signaling network."

The data presented here shows that a partial PolIII CTD mutant in Ser5, with compromised initiation, displays a defect on ESR induction/repression. Unfortunately, the authors do not prove that Ser5 of the CTD is directly phosphorylated by the MAPK, as it could be interpreted by their claim. They do not show that this Ser5 Phosphorylation is mediated by Hog1 *in vivo* and that this is the mechanism for ESR regulation. It is also not clear that this effect by the MAPK is critical for the differential ESR regulation. It is known that Hog1 participates in transcription initiation and thus, mutants with compromised initiation, such as the partial Ser5 mutant of the PolIII, should affect regulation of the ESR genes. The data presented here still can be just an indirect effect of the alteration of the transcriptional process rather than a direct effect by the MAPK. Thus, this result is expected and confirms previous data from many laboratories, but does not provide any additional insight on how the MAP could be acting on Pol II regulation.

3rd Revision - authors' response

03 October 2014

We have made your final requested changes by toning down the language about the *in vivo* role of Hog1 in direct CTD modification, in the following places:

1. We removed in its entirety on Page 8 the sentence, "Together, these results suggest that Hog1 directly phosphorylates the Rpb1-CTD in response to stress."
2. We toned down the language in several places on page 8 with regard to CTD coordination of ESR modules. In several other places, we changed the language from "direct modification of Rpb1-CTD" to "regulation of Rpb1-CTD."
3. In the Discussion, we changed the statement "Hog1 dictates [Rpb1-CTD] modification *in vivo*" and change to "Hog1 is required for normal modification of Rpb1-CTD."
4. In the Discussion section, where we feel it is appropriate to put forward a model based on our results, we changed the final statement into one of a model with the word 'perhaps': "We propose that direct regulation of RNA Pol II, perhaps in part by the Hog1 kinase, plays a central role in coordinating these opposing transcriptional modules."

We have uploaded all the mass spectrometric proteomic data into an accepted public repository. We also provide several datasets that can be linked directly to the relevant figures.